# SPKLIP: Aligning Spike Video Streams with Natural Language

## Abstract

Spike cameras offer unique sensing capabilities but their sparse, asynchronous output challenges semantic understanding, especially for Spike Video-Language Alignment (Spike-VLA) where models like CLIP underperform due to modality mismatch. We introduce SPKLIP, the first architecture specifically for Spike-VLA. SPKLIP employs a hierarchical spike feature extractor that adaptively models multi-scale temporal dynamics in event streams, and uses spike-text contrastive learning to directly align spike video with language, enabling effective few-shot learning. A full-spiking visual encoder variant, integrating SNN components into our pipeline, demonstrates enhanced energy efficiency. Experiments show state-of-the-art performance on benchmark spike datasets and strong few-shot generalization on a newly contributed real-world dataset. SPKLIP's energy efficiency highlights its potential for neuromorphic deployment, advancing event-based multimodal research. The source code and dataset are available at [link removed for anonymity].

## 1 Introduction

Inspired by retina, spike cameras (Huang et al., 2023) represent a paradigm shift for high-speed motion perception, capable of operating at effective frame rates up to 40,000 Hz with an exceptional dynamic range around 180 dB. This unique combination makes them ideal for capturing complex, rapid dynamics often missed by conventional cameras. However, translating this raw sensing potential into high-level semantic understanding remains a significant hurdle. Current approaches often resort to converting the native, sparse spike event streams into static, image-like representations (Zhao et al., 2021b; Fan et al., 2024a; Zhang et al., 2024b; Chen et al., 2025; Wang et al., 2021; Liang et al., 2023; Ercan et al., 2023; Rudnev et al., 2023). This simplification, while sometimes useful for basic recognition, inadvertently discards the rich, continuous spatiotemporal information crucial for interpreting fast-evolving actions and events – essential data for real-time applications like autonomous navigation, robotic interaction, or high-speed quality control (Nahavandi et al., 2022; Robinson et al., 2023).

Furthermore, the remarkable progress achieved by vision-language models like CLIP (Radford et al., 2021b) in grounding semantics for standard RGB videos (Ma et al., 2022; Wasim et al., 2023; Wang et al., 2024c; Luo et al., 2022; Wang et al., 2024b; Tang et al., 2021) does not readily transfer to the spike domain. These powerful models suffer severe performance degradation when applied directly due to the fundamental mismatch between their dense, synchronous frame processing assumptions and the asynchronous, event-driven nature of spike data. This incompatibility prevents the direct leveraging of state-of-the-art (SOTA) semantic alignment techniques for advanced spike-based perception, leaving a critical gap in our ability to interpret these information-rich data streams linguistically. Bridging this gap necessitates overcoming challenges unique to spike video analysis: specialized feature extraction for sparse, asynchronous data (Zhao et al., 2023; Xia et al., 2023; Gallego et al., 2020; Messikommer et al., 2025; Dong et al., 2024; Feng et al., 2024; Zhang et al., 2024a; Su et al., 2024; Zhao et al., 2024; Zhu et al., 2024), data scarcity for labeled spike videos (Farchy et al., 2013; Lund & Miglino, 1996; Koos et al., 2010; Carpin et al., 2007; Koopman et al., 2024), and the need for algorithmic efficiency in power-constrained scenarios (Menghani, 2023; Tay et al., 2022).

To address these multifaceted challenges and unlock the potential of spike cameras for high-level scene understanding, we introduce **SPKLIP** (Spike-based Cross-modal Learning with CLIP). To

our knowledge, SPKLIP is the first neural network architecture specifically designed for Spike Video-Language Alignment (Spike-VLA). SPKLIP aims to achieve robust semantic interpretation of high-speed dynamic scenes directly from spike event streams through multimodal contrastive learning, explicitly tackling the limitations of prior work. Alongside algorithmic innovations, we contribute a new real-world spike video dataset to foster research under realistic conditions.

Our core contributions are:

- **A Novel Spike-VLA Architecture:** We introduce SPKLIP, the first end-to-end framework for Spike Video-Language Alignment. It features a hierarchical spike feature extractor (HSFE) specifically designed for sparse, high-frequency spike data streams—unlike conventional extractors—and employs Spike-Text Contrastive Learning (STCL) to directly align raw spike video with text, bypassing intermediate frame conversion.
- **Energy-Performance Trade-off Analysis and Real-World Validation:** We develop a Full-Spiking Visual Encoder (FSVE) as an exploratory study integrating SNN principles, providing the first analysis of the complex trade-offs between energy efficiency and performance for the Spike-VLA task. Furthermore, SPKLIP's effectiveness and generalization, including few-shot learning, are validated on **a newly contributed real-world spike video dataset, which we also release to the community.**
- **Establishing a Strong Baseline:** Through comprehensive experiments, SPKLIP is shown to significantly outperform adapted conventional vision-language models on spike-VLA.

## 2 RELATED WORK

Video action recognition has evolved significantly. Early approaches often relied on handcrafted spatiotemporal features, such as HOG and MBH (Dalal & Triggs, 2005; Laptev et al., 2008; Wang & Schmid, 2013; Zhu et al., 2016b;a), combined with classifiers like SVMs (Cortes & Vapnik, 1995; Schölkopf et al., 1998). Subsequently, deep learning frameworks, including 3D CNNs (Tran et al., 2015; Noor & Park, 2023; Wang et al., 2024a), SlowFast networks (Feichtenhofer et al., 2019; Dai et al., 2023; Bae et al., 2024), and Temporal Shift Modules (TSM) (Lin et al., 2019), achieved substantial performance gains by effectively modeling temporal correlations within sequences of dense frames. However, the computational demands and reliance on dense video data associated with these methods have motivated exploration into alternative sensing modalities. Event cameras and spike cameras have emerged as promising alternatives, offering benefits like low power consumption, high dynamic range, and high temporal resolution sensing. Research in this area has explored various ways to utilize these sensors. For instance, some works focus on fusing data from conventional cameras with event streams using Transformers and Spiking Neural Networks (SNNs) (Wang et al., 2023; Fan et al., 2024b; Hwang et al., 2024; Zhou et al., 2024b; Ren et al., 2023; Yao et al., 2023; Gao et al., 2025). Others have integrated event features with semantic priors via multimodal Transformers (Li et al., 2023; Zhou et al., 2024a; Kong et al., 2024; Li et al., 2025). Processing spike data effectively involves addressing its unique characteristics, such as signal sparsity and noise patterns. Aligning these unique event streams directly with textual semantics presents an interesting avenue for further research. Recent advancements have also focused on enhancing action recognition by integrating textual information with visual data. Techniques include using large language models (LLMs) to enrich action semantics from spatiotemporal descriptors (Chen et al., 2024; Wang et al., 2024d) and generating video-conditional text embeddings (Kahatapitiya et al., 2024). These studies highlight the value of multimodal approaches, often involving fusion strategies between text and RGB or event data representations.

## 3 METHODOLOGY

We propose a hybrid architecture, SPKLIP, which learns joint representations from spike video streams and raw text tokens, enabling end-to-end learning. The main architecture of SPKLIP, illustrated in Fig. 1, is to enhance the ability of the visual encoder to extract spike modality features. More specifically, a dedicated Hierarchical Spike Feature Extractor (HSFE) is constructed, addressing the challenges posed by the sparse and asynchronous nature of spike data (Fig. 1a). Also, a hierarchical feature fusion module is used to align closely with textual descriptions, enabling applications in various downstream tasks such as video question answering and text-to-video retrieval.

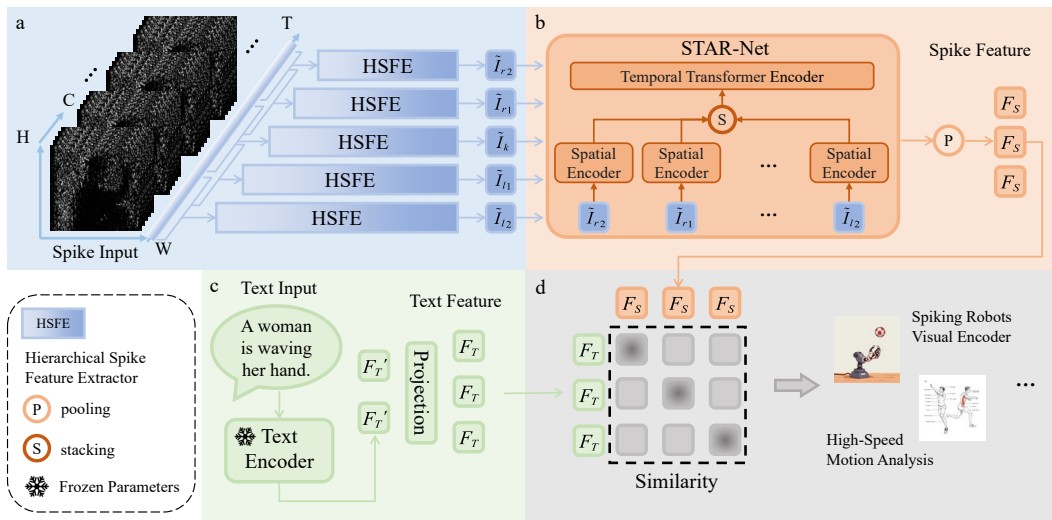

Figure 1: Illustration of the proposed end-to-end Spike-Based Video Understanding Framework (SPKLIP). This framework primarily consists of four key components: the Hierarchical Spike Feature Extractor (HSFE), the SpatioTemporal Attentive Residual Network (STAR-Net) module, a Text Encoder, and a Contrastive Learning Framework. Each component plays a critical role in enabling robust and efficient video understanding.

## 3.1 SPIKE CAMERA

Spike cameras are inspired by the sampling principle of retina fovea, which consists of an array of pixels, each of which continuously accumulates incident light intensity $I(t)$. When the accumulated charge reaches a predefined threshold $\theta$, the pixel fires a spike signal (i.e., a "pulse") and resets the integrator to initiate a new "integrate-and-fire" cycle. Under this mechanism, the instantaneous charge $A(t)$ on the integrator is formulated as:

$$A(t) = \left( \int_0^t \alpha \cdot I(x) \, dx \right) \bmod \theta, \tag{1}$$

where $\alpha$ represents the photoelectric conversion rate. Ideally, spikes can be triggered at arbitrary time instants $t_k$, satisfying: $\int_0^{t_k} \alpha \cdot I(x) \, dx = k\theta$, which implies $A(t_k) = 0$, with $k$ denoting the spike index. However, constrained by circuit limitations, spike detection must be discretized. Pixels output spikes as discrete-time signals $S(n)$, where spike flags are periodically checked at intervals $t = nT$ ($n = 1, 2, \dots$), with $T$ being a microsecond-scale interval. Specifically: If a spike flag is detected at $t = nT$, $S(n) = 1$ is recorded, and the flag is reset to prepare for the next spike. Otherwise, $S(n) = 0$ is recorded. Under continuous light exposure, all pixels on the sensor operate simultaneously and independently, firing spikes to encode photon arrivals. The sensor employs high-speed polling to inspect the binary spike status ("0" or "1") of each pixel, generating an H × W spike frame. Over time, the camera outputs a sequence of such frames, forming an H × W × N binary spike stream $S(x, y, n)$. Detailed principles of spike camera can be found in Appendix A.1.

## 3.2 HIERARCHICAL SPIKE FEATURE EXTRACTOR (HSFE)

HSFE comprises two key components: Multi-Scale Temporal Filtering (MTF) and Spatial Attention (SA). MTF balances noise suppression and motion detail preservation. Fixed-time window methods struggle to reconcile noise suppression with motion detail preservation in asynchronous, sparse spike streams (Zhao et al., 2021a). To address this, MTF adaptively models temporal dynamics at varying scales. The input spike stream [B, T, C, H, W] is first reshaped into [T × C, H, W] and divided into five temporally overlapping sub-blocks via a sliding window (radius=30, step=45). Each sub-block centers on a key time step, defined as:

$$B_{\text{block}_i} = S\left[t_i - \text{r}_{\text{win}} : t_i + \text{r}_{\text{win}} + 1\right], \tag{2}$$

where $S$ is the original stream and $r_{\text{win}}$ is the window radius.

Multi-scale convolutional branches extract features with adaptive temporal resolutions. Each sub-block is processed in parallel using convolutional kernels with varying input channel dimensions. Reducing channel count broadens temporal coverage (simulating longer "virtual exposure time") but sacrifices fine-grained details, while increasing channels focuses on short-time high-frequency features (e.g., rapid motion). A learnable temporal mask $M_i \in \mathbb{R}^{1 \times 1 \times N}$ dynamically weights spikes via element-wise multiplication: $H_t^{(i)} = \text{Conv}_{k_i}(M_i \circ B_{\text{block}_i})$, where $k_i$ denotes channel size for branch $i$.

Photon conservation governs multi-branch channel allocation. The total photon quantity within each spike cycle is physically constrained by the camera's trigger mechanism:

$$\text{Photon total} = \theta \cdot |\phi_n| \cdot \sum_{i \in \phi_n} S_i(x, y),$$

$$k_i \propto \frac{\text{Photon total}}{T_i}. \tag{3}$$

Here, $\theta$ is the threshold, $\phi_n$ denotes the virtual exposure window, and $S_i(x, y)$ is the binary spike signal. This constraint ensures that larger $k_i$ (higher channel counts) reduce temporal coverage $T_i$ for high-frequency motion capture, while smaller $k_i$ extend $T_i$ to stabilize static regions. This design follows a fluid-container analogy: fixing $\text{Photon\_total}$, increasing base area ($k_i$) reduces height ($T_i$), and vice versa.

SA enhances critical time steps and suppresses noise. An attention module $a(\cdot)$ learns modulation weights to prioritize relevant temporal scales: $[W_t^{(1)}, \ldots, W_t^{(m)}] = a([H_t^{(1)}, \ldots, H_t^{(m)}])$. The output is a stacked feature map: $\tilde{I}_t = [W_t^{(1)} \circ H_t^{(1)}, \ldots, W_t^{(m)} \circ H_t^{(m)}]$. Here, $m$ is the branch count, and $\circ$ denotes element-wise multiplication. The module applies MTF and SA to five adjacent spike blocks $\{B_{l2}, B_{l1}, B_k, B_{r1}, B_{r2}\}$, generating coarse estimates $\{\tilde{I}_{l2}, \tilde{I}_{l1}, \tilde{I}_k, \tilde{I}_{r1}, \tilde{I}_{r2}\}$ that describe instantaneous intensity characteristics across time steps, jointly modeling short-term temporal dependencies.

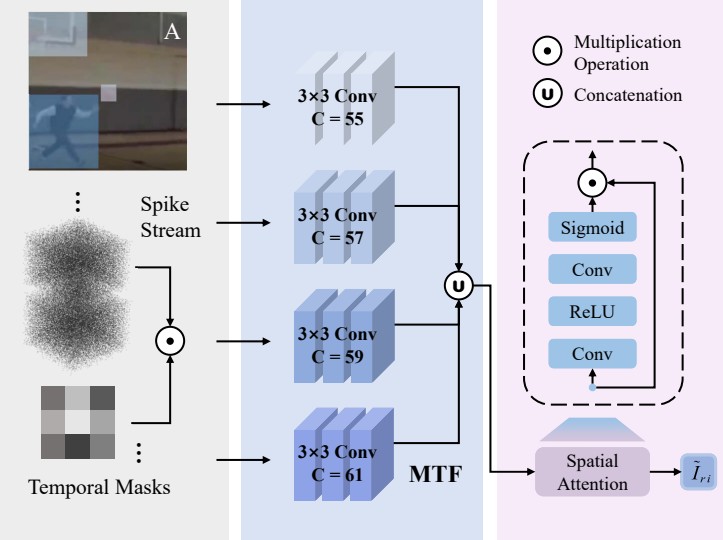

Figure 2: This figure illustrates the mechanism of the HSFE module. By employing a Temporal Mask and convolutions with varying channel sizes, the HSFE adaptively balances noise suppression with motion preservation. For example, in A, features are extracted at different temporal scales corresponding to the fast-moving basketball, the medium-speed person, and the static basketball hoop. The resulting features are then concatenated and processed by a Spatial Attention module, which computes weights to enhance the contribution of the most informative temporal steps.

### 3.3 SPATIOTEMPORAL ATTENTIVE RESIDUAL NETWORK (STAR-NET)

The coarse-grained instantaneous light intensity features $\tilde{I}_{l2}, \tilde{I}_{l1}, \tilde{I}_k, \tilde{I}_{r1}, \tilde{I}_{r2}$ output by HSFE are processed through a two-stage fusion module to model long-range spatiotemporal dependencies: MAPResNet and Transformer. MAPResNet enables hierarchical feature extraction with hybrid attention. As the backbone network, MAPResNet (Modified Attention-Pooling ResNet), integrates CNNs and global attention for multi-scale feature learning. It follows a hierarchical design with three components: (1) A stem module with three stacked convolutions (3×3 kernels, stride=2) for initial feature extraction; (2) Four residual block groups (with 2, 2, 2, 2 bottleneck blocks) progressively expanding channel dimensions from 64 to 2048 via 4× expansion ratios; (3) An attention pooling module applying multi-head self-attention ($h = 8$) over flattened spatial tokens ($\frac{H}{32} \times \frac{W}{32}$) with learnable positional encodings. This hybrid CNN-transformer architecture combines local feature extraction (via residual bottlenecks (He et al., 2015)) with global attention pooling, following recent paradigms (Vaswani et al., 2023). Input features $\tilde{I}_{l2}, \tilde{I}_{l1}, \tilde{I}_k, \tilde{I}_{r1}, \tilde{I}_{r2}$ are first processed by the stem module, then refined through residual blocks, and finally compressed into high-level representations $[B, D]$ via attention pooling. This extends attention-pooling strategies in vision-language pretraining (Radford et al., 2021a).

Transformer-based temporal fusion models long-range dependencies. A Transformer encoder captures cross-frame relationships in the time series. Features from MAPResNet are stacked along the temporal dimension as $[T, B, D]$, then processed by multi-head self-attention:

$$\text{Attention}(Q, K, V) = \text{softmax}\left(\frac{QK^T}{\sqrt{d_k}}\right) V. \tag{4}$$

The output retains shape $[T, B, D]$, now encoding temporal context. Finally, global feature pooling averages across time:

$$\text{global feature} = \frac{1}{T} \sum_{t=1}^{T} \text{temporal features}[t, :], \tag{5}$$

producing a compact representation $F_s \in [B, D]$, as illustrated in Fig. 1b.

### 3.4 SPIKE-TEXT CONTRASTIVE LEARNING (STCL)

STAR-Net extracts unified embeddings for spike-based videos and natural language texts, enabling cross-modal alignment via contrastive learning. Text encoder maps language tokens into a shared semantic space.

The text encoder follows the BERT architecture (Devlin et al., 2019), converting discrete text tokens into continuous embeddings. Specifically: (1) Input tokens are mapped to vectors via a learnable token embedding layer; (2) Positional encodings are added to preserve sequential context; (3) A Transformer encoder captures contextual dependencies; (4) Output features are projected through a 'text projection' layer to align with the visual embedding space (Fig. 1c).

Contrastive loss maximizes inter-modal similarity and intra-modal discrimination. Given video embeddings $v_i \in [B, \text{embed\_dim}]$ and text embeddings $t_i \in [B, \text{embed\_dim}]$, the objective is to align positive pairs while separating negatives:

$$\mathcal{L} = -\frac{1}{B} \sum_{i=1}^{B} \left[ \log \frac{\exp\left(\text{sim}(v_i, t_i)/\tau\right)}{\sum_{j=1}^{B} \exp\left(\text{sim}(v_i, t_j)/\tau\right)} + \log \frac{\exp\left(\text{sim}(t_i, v_i)/\tau\right)}{\sum_{j=1}^{B} \exp\left(\text{sim}(t_i, v_j)/\tau\right)} \right]. \tag{6}$$

Here $B$ is batch size, $\text{sim}(v, t)$ cosine similarity between $v$ and $t$, and $\tau$ learnable temperature parameter (`logit_scale`) controlling similarity distribution smoothness. This symmetric loss formulation ensures mutual alignment: videos are attracted to matched texts and repelled by mismatches, and vice versa.

### 3.5 FULL-SPIKING VISUAL ENCODER (FSVE)

We propose a pure spiking visual encoder (FSVE) that integrates Spiking ResNets with a Spiking Temporal Transformer for event stream processing. The architecture combines leaky integrate-and-fire neurons with temporal-dependent normalization for stable spatial feature extraction, and

a spike-driven self-attention mechanism enabling energy-efficient spatiotemporal modeling. This co-design achieves end-to-end spike-domain computation while preserving biological plausibility. See Fig. 3 and Appendix A.5 for details.

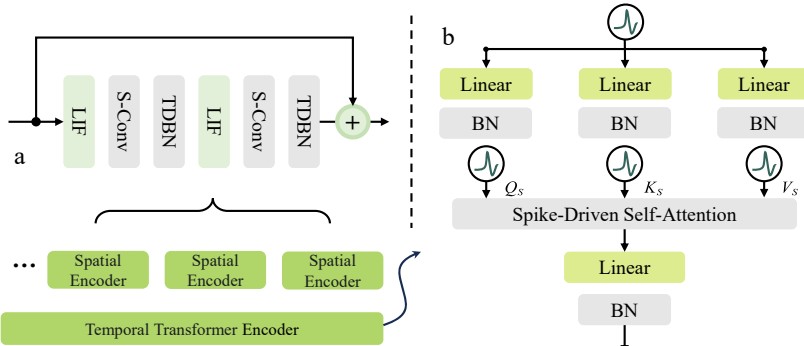

Figure 3: Architecture overview of FSVE. (a) Spiking ResNets extract spatial features with LIF neurons and TDBN (Hu et al., 2021). (b) E-SDSA module (Yao et al., 2025b) implements spike-driven attention with threshold normalization and sparse computation.

## 4 EXPERIMENT

### 4.1 EXPERIMENTAL SETTINGS

**Datasets**    We employed HMDB51-S, UCF101-S, and a custom dataset as primary experimental data. The first two datasets were generated by converting the renowned HMDB51 and UCF101 datasets using the SpikeCV toolkit (Zheng et al., 2024), preserving most characteristics of the spike modality. The self-built dataset comprises 30 action categories (e.g., badminton bat swings, table tennis forehand loops, and other high-speed, high-dynamic movements) captured in real-world scenarios using a spike camera. HMDB51-S contains 51 action categories with 6,849 spike videos, while UCF101-S consists of 101 action categories encompassing 13,320 spike videos. All videos maintain a resolution of 320×240 pixels, with frame counts varying between 2,000 and 4,000 frames.

**Real Dataset Preprocessing Pipeline**    In the real dataset acquisition and processing pipeline, we used a spike camera to obtain data at an original resolution of $416 \times 250$. For model compatibility, we first performed center cropping to adjust the frame size to $320 \times 240$. To maximize sample diversity and enhance model generalizability, continuous long videos for each action category were captured by multiple, distinct individuals. Through this rigorous pre-processing of the long videos, we ultimately yielded a robust dataset comprising $96 \times 30$ samples (96 samples per category $\times$ 30 categories) for subsequent evaluation.

**Implementation**    Since this work proposes the first architecture of its kind, the visual encoder in our model was trained from scratch without utilizing any pretrained weights. The training configuration employed a batch size of 8 over 30 epochs with a learning rate of 2e-5, optimized by the AdamW algorithm. Our model directly processes spike-modality data without requiring any reconstruction preprocessing. The framework was implemented using PyTorch and trained on NVIDIA A100 GPUs.

### 4.2 COMPARATIVE ANALYSIS OF VIDEO-CLIPS AND SPKLIP

Methods designed for RGB modality underperform on spike data, while SPKLIP achieves SOTA results. As shown in Table 1, we compare state-of-the-art visual encoders for video-based spike data semantic understanding. The table is structured into three parts:

(1) Top 4 rows: RGB-based methods (X-CLIP, Vita-CLIP, MotionPrompt, OmniCLIP) evaluated on HMDB51 with CLIP-400M pretrained weights (Liu et al., 2024). (2) Middle 2 rows: RGB-based methods (M2-CLIP, Vita-CLIP), adapted to spike modality by input dimension adjustments while

retaining original architectures. Details are provided in the Appendix A.3. (3) Bottom row: Our SPKLIP model for spike modality with ResNet-18 backbone trained from scratch.

All datasets maintain 240×320 resolution. After 30 epochs, we evaluate Top-1/Top-5 accuracy using official learning rates and optimizers. This structured comparison highlights the performance gap between RGB and spike modality methods.

Table 1: Comparison of Top-1/Top-5 accuracy between SPKLIP and SOTA RGB/Spike-based methods on HMDB51(-S) datasets. (+A indicates that the model has been adapted)

| Type | Method | Reference | Pre-trained | ACC | | Dataset |
|------|--------|-----------|-------------|-----------|-----------|---------|
| | | | | Top-1 (%) | Top-5 (%) | |
| RGB | X-CLIP | ECCV-2022 | CLIP-400M | 70.94 | 93.39 | HMDB51 |
| | Vita-CLIP | CVPR-2023 | CLIP-400M | 71.18 | 94.12 | HMDB51 |
| | MotionPrompt | ACM MM-2023 | CLIP-400M | 72.89 | 93.21 | HMDB51 |
| | OmniCLIP | ECAI-2024 | CLIP-400M | 76.64 | 95.89 | HMDB51 |
| Spike | M2-CLIP (A) | AAAI-2024 | - | 39.57 | 85.96 | HMDB51-S |
| | Vita-CLIP (A) | CVPR-2023 | - | 45.31 | 87.14 | HMDB51-S |
| | **SPKLIP (ours)** | - | - | **91.15** | **99.75** | HMDB51-S |

Our results underscore the necessity of a specialized architectural design for the spike modality. To establish a fair comparison, we meticulously adapted prominent conventional models, including M2-CLIP and Vita-CLIP, to process the spike data. Despite this direct adaptation, their performance on the HMDB51-S dataset collapsed, as evidenced in Table 1. This confirms that, despite being adapted for spike inputs, conventional architectures (e.g. Vision Transformers), which are optimized for dense pixels, fundamentally struggle with the sparse, event-driven nature of spike streams.

In stark contrast, our SPKLIP, an architecture natively designed for this modality, achieves a superior 91.15% Top-1 accuracy on the same task. This performance gap is not merely an incremental improvement but a demonstration of SPKLIP's valid spatiotemporal feature extraction framework, establishing a critical and robust new benchmark for spike-based vision.

### 4.3 EVALUATE WITH DATA FROM REAL SHOTS

The Sim-to-Real Domain Gap as a Challenge. To validate our model's generalization, we first quantify the significant sim-to-real domain gap. Our analysis confirms both a key similarity and two significant differences:

Fundamental Similarity: Both domains are fundamentally **sparse**. The vast majority (99%) of pixel activity in both distributions is concentrated in the low-count 0-30 spike range, providing a common sparse foundation.

Significant Differences: (1) **Motion Statistics:** The real data represents "sparse, local motion" (e.g., clapping, mean activity 0.0179), while the synthetic data represents "dense, global motion" (mean 0.0576). (2) **Artifact Patterns:** Each domain contains unique, high-intensity noise; real data shows sensor-specific artifacts (e.g., peak ∼60 horizontal lines), while synthetic data shows algorithmic artifacts (e.g., peak ∼30 background blocks).

A detailed breakdown of this analysis, including quantitative charts and qualitative heatmaps (see Fig. A1-1, 2, 3), is provided in Appendix A.2. This substantial gap in both motion statistics and noise patterns makes sim-to-real transfer a non-trivial challenge. We therefore adopt a few-shot adaptation approach to validate our pre-trained model's ability to cross this gap and generalize to real-world spike streams.

Few-shot adaptation validates simulation-to-reality generalization. We evaluate our model's performance on a self-collected, real-world dataset. Due to the domain gap between physical spike cameras and simulated environments, we adopt a few-shot adaptation approach: most model parameters

remain frozen, with only the final two layers of STAR-Net fine-tuned. As shown in Fig. 4, we test 2-shot, 4-shot, 6-shot, and 8-shot settings to assess generalization.

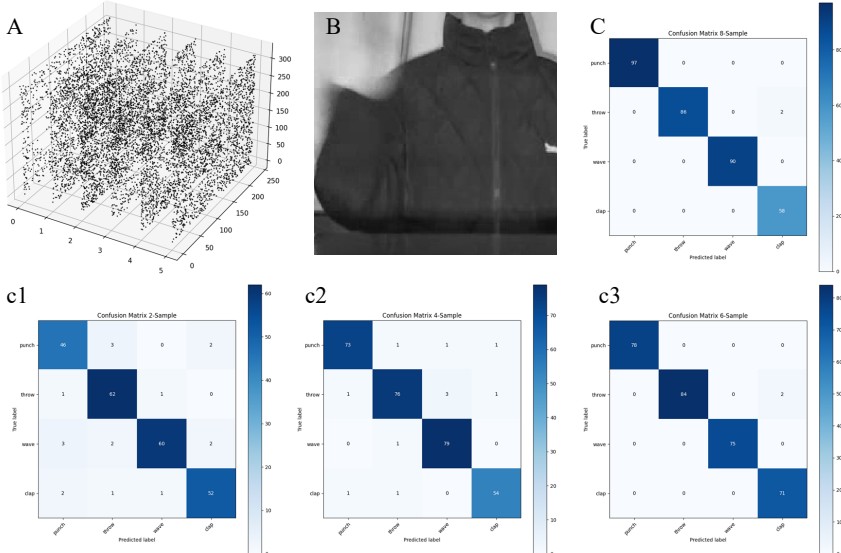

Figure 4: Performance Evaluation on Real Spike Camera Data: (A) 3D visualization of raw spike stream; (B) Processed video (wave); (C) Confusion matrix, which displays a subset of 4 high-speed action categories from the total 30 classes for clarity. Top-1 accuracy: 55.17% (2 shots), 73.11% (4 shots), 85.78% (6 shots), 88.12% (8 shots).

Performance improves consistently with increased shot counts. Results show progressive improvement as shot counts increase: (1) 2 shots: 55.17% Top-1 accuracy (limited adaptation capacity); (2) 4 shots: 73.11% (+17.94%), demonstrating rapid learning with minimal data; (3) 6 shots: 85.78% (+12.67%), approaching full-dataset performance; (4) 8 shots: 88.12% (+2.34%), achieving near-optimal accuracy.

This trend highlights the framework's robust simulation-to-reality generalization, with minimal fine-tuning required for real-world deployment.

### 4.4 ABLATION STUDY OF PROPOSED METHOD

Key components contribute progressively to model performance. We conduct ablation studies to analyze the impact of individual components (MTF, SA, STAR-Net) on UCF101-S and HMDB51-S datasets. The specific dataset transformation construction method is presented in detail in A.8. All experiments use ResNet-18 as the backbone and 250 input frames per spike video unless specified otherwise. Table 2 and Table 4 summarize results.

To evaluate the contribution of Photon conservation (equation 3) (which implements dynamic channel slicing selection for early feature extraction branches through the channel_step parameter), we conducted an ablation experiment in Table 2. In the full model, the parallel convolutional branches in HSFE enable simultaneous feature capture of both high-frequency rapid motion and low-frequency stable regions. For the ablated model, we removed this channel slicing mechanism. Specifically, all parallel convolutional branches in HSFE received and processed complete input feature maps, with their respective input channels adjusted to the full count during initialization.

As evidenced by the results in Table 2, restricting the core functionality of HSFE leads to a 2.21% degradation in Top-1 accuracy on HMDB51-S compared to the complete SPKLIP model. This performance gap demonstrates a substantial impact, conclusively validating the superior capability of the HSFE module.

Table 2: Ablation study demonstrating the contribution of the HSFE.

| Model Configuration | Dataset | ACC(%) Top-1 |
|---|---|---|
| HSFE (Ablation) | HMDB51-S | 88.94 |
| HSFE (Full Model) | HMDB51-S | **91.15** |

We also conducted empirical studies on the number of temporal sub-blocks (which the HSFE processes) to find an optimal balance. We report the Top-1 accuracy on HMDB51-S for 3, 5 (our final model), and 7 sub-blocks.

Table 3: Ablation study on the number of HSFE temporal sub-blocks.

| Number of Sub-blocks | Top-1 Accuracy (HMDB51-S) |
|---|---|
| 3 Blocks | 88.56% |
| **5 Blocks (Ours)** | **91.15%** |
| 7 Blocks | 90.46% |

As shown in Table 3, the results clearly demonstrate an empirical trade-off: (1) **Too Few (3 blocks):** Performance degraded significantly by 2.59%. This suggests that 3 sub-blocks are insufficient to capture the full temporal context of actions in HMDB51-S. (2) **Too Many (7 blocks):** Performance also degraded (by 0.69%). We attribute this to "Temporal Context Pollution," where 7 sub-blocks cover an excessively long time window, forcibly introducing irrelevant "noise" (e.g., static states before or after the core action), which is unsuitable for the characteristics of this dataset. Therefore, 'five sub-blocks' is not an absolute hyperparameter, but rather the optimal empirical balance we found for this benchmark, balancing 'insufficient context' against 'context pollution'.

Table 4: Ablation study demonstrating the contribution of key components (MTF, SA, STAR-Net) to Top-1 accuracy on UCF101-S and HMDB51-S. The value is shown in the format of mean±standard deviation, calculated across 5 trials.

| Components | | | ACC(%) Top-1 | |
|---|---|---|---|---|
| MTF | SA | STAR-Net | UCF101-S | HMDB51-S |
| ✓ | ✗ | ✗ | $76.19 \pm 0.46$ | $80.80 \pm 2.23$ |
| ✓ | ✓ | ✗ | $77.64 \pm 0.44$ | $82.42 \pm 1.84$ |
| ✓ | ✓ | ✓ | $\mathbf{86.43} \pm 0.32$ | $\mathbf{91.15} \pm 2.21$ |

MTF and SA improve spatial-temporal feature learning; STAR-Net enhances global context. We split the HSFE module into two components, Multi-Scale Temporal Filtering (MTF) and Spatial Attention (SA), and test their importance separately. Table 4 decomposes the contributions of MTF, SA, and STAR-Net. (1) MTF: The limited performance of general-purpose models like M2CLIP when directly applied to spike data (as shown in Table 1) highlights the limitations of unspecialized temporal filtering. In contrast, our MTF module alone (Table 4: 76.19% on UCF101-S, 80.80% on HMDB51-S) effectively captures crucial motion details, validating the necessity of a tailored approach for spike-based inputs. (2) SA: Adding SA to MTF further enhances spatial feature extraction, achieving 1.45% and 1.62% gains. (3) STAR-Net: Integrating STAR-Net's dual-stage spatiotemporal fusion mechanism boosts performance by 8.79% (UCF101-S) and 9.73% (HMDB51-S), demonstrating its ability to model complex long-range dependencies. These results validate the incremental improvements from each component, confirming their collaborative role in advancing spike-modality action recognition.

## 4.5 VISUALIZATION OF TEMPORAL DYNAMICS

To analyse the internal temporal dynamics of our model, we conducted a visualisation experiment. To truly isolate the dynamic features, we use PyTorch Hooks to extract the five feature vectors $(V_1...V_5)$ from the HSFE module. We then subtract the mean vector, $V_{\text{avg}}$ to remove the static components. Finally, we compute the cosine correlation heatmap of these five "dynamic-only" vectors, as shown in Fig. 5.

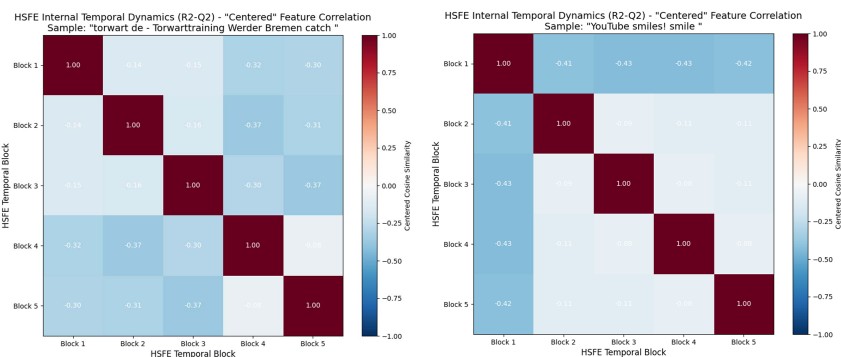

Figure 5: Visualization of HSFE's internal temporal dynamics using "centered" feature correlation heatmaps.

As shown in Fig. 5, the heatmaps are not flat (i.e., not monochromatic) but exhibit highly structured, content-dependent patterns. For instance, in the "smile" sample (right), the "neutral" feature (Block 1) is strongly **negatively correlated** ($\sim$-0.4, blue) with the subsequent "smiling" features (Blocks 2-5). In contrast, the "catch" sample (left) shows a completely different complex pattern. This significant "variation" in correlation and the "pattern difference" across samples are definitive proof that our HSFE module is not a static processor; it is **dynamically extracting distinct, time-varying features** based on the video content.

## 4.6 VALIDATING MULTIMODAL ALIGNMENT VIA TEXT-TO-VIDEO RETRIEVAL

To quantitatively substantiate that our framework learns a meaningful joint embedding space, we conducted a rigorous text-to-spike-video retrieval task, moving beyond a simple classification mission.

The retrieval performance, detailed in Table 5, confirms the model's strong alignment capabilities. These results provide direct empirical evidence for the effectiveness of our cross-modal learning strategy, validating that SPKLIP successfully maps sparse spike streams and natural language into a shared, semantically coherent space. A detailed description of the implementation is available in Appendix A.4.

Table 5: Text-to-Video Retrieval Performance on Spike Datasets.

| Datasets | Recall@1 (R@1) | Recall@5 (R@5) | Recall@10 (R@10) |
|---|---|---|---|
| HMDB51-S | 31.94% | 63.12% | 75.10% |

## 5 CONCLUSION

This work introduced SPKLIP, the first architecture for Spike Video-Language Alignment (Spike-VLA). Using a specialized Hierarchical Spike Feature Extractor and Spike-Text Contrastive Learning, SPKLIP significantly outperformed adapted conventional models on benchmark spike datasets and demonstrated effective few-shot learning on a new real-world dataset. Our full-spiking variant also highlights a path towards energy-efficient semantic perception. SPKLIP provides a foundational framework for advancing multimodal tasks with event-based data on neuromorphic platforms.

## DECLARATION OF LLM USAGE

During the preparation of this manuscript, we used a Large Language Model (LLM) for assistance. We only used the LLM to improve the clarity and readability of the text, which included correcting the grammar, checking the spelling, and translating the text into English.

## ETHICS STATEMENT

The research presented in this paper adheres to the highest ethical standards. The primary datasets used, HMDB51-S and UCF101-S, are derived from publicly available, widely used academic benchmarks (HMDB51 and UCF101) for action recognition, which do not contain personally identifiable or sensitive information. Our custom-collected real-world dataset consists of anonymized recordings of common human actions (e.g., clapping, waving) performed by consenting participants in non-public settings. The work is foundational in nature, aiming to advance the scientific understanding of spike-based vision, and we do not foresee any direct negative societal impacts or potential for misuse.

## REPRODUCIBILITY STATEMENT

To ensure the reproducibility of our results, we commit to making our source code, including the model architecture, training scripts, and evaluation protocols, publicly available upon publication. Our experiments were conducted using PyTorch on NVIDIA A100 GPUs. The HMDB51-S and UCF101-S datasets were generated using the publicly available SpikeCV toolkit, and the conversion process is detailed in the appendix. All hyperparameters, such as learning rate, batch size, and optimizer details, are explicitly stated in section 4. The custom-collected dataset will also be released to facilitate further research in the community.

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

# A APPENDIX

## A.1 IMPLEMENTATION DETAILS OF THE SPIKE CAMERA

### A.1.1 IMAGING MECHANISM AND OUTPUT FORMAT

A **spike camera** is a bio-inspired sensor where each pixel independently accumulates incoming photons. When the photon count at a pixel hits a fixed threshold, it fires a binary '1' spike and instantly resets. This means **light intensity is encoded by the spike frequency**.

The raw hardware output is a series of binary snapshots ($H \times W$ matrices of 1s and 0s) captured at an extremely high frequency (e.g., 40,000 Hz). The full output is therefore a 3D binary data stream, $S(x, y, t)$, where $(x, y)$ are pixel coordinates and $t$ is the discrete time step.

### A.1.2 DEFINITION OF A "FRAME" AND TEMPORAL INFORMATION

While the individual spike *timings* at each pixel is asynchronous (since they depend on light intensity), the data *readout* is synchronous—all pixels are sampled simultaneously at a very high rate.

So, a "frame" in this context is simply **one of these high-frequency binary snapshots**, not a conventional intensity image. The crucial **temporal information** is captured in the sequence of these frames. By counting spikes over a time window, we can reconstruct brightness. By measuring the time *between* spikes (Inter-Spike Interval), we can infer instantaneous changes in brightness. This allows the camera to capture high-speed dynamics that traditional cameras miss due to motion blur.

## A.2 AN ANALYSIS ABOUT THE SIM-TO-REAL DOMAIN GAP.

**1. Fundamental Similarity (Sparsity):** As shown in Fig. A1 (Plot 1), both datasets share the fundamental physical property of spike signals: high sparsity. Although their means differ, the vast majority (99%) of all pixel activity in both distributions is concentrated in the low-count range (0-30 spikes per pixel). This shared sparsity provides a common foundation for our model.

**2. Significant Domain Gap (Statistics & Artifacts):** However, built upon this shared sparse foundation, the two datasets diverge significantly:

(a) Motion Statistics (Local vs. Global): The statistics (Fig. A1, Plot 1 & 2) differ because the nature of the motion is different. The synthetic data (orange) is statistically denser (mean 14.77) and "bursty" (mean 0.0576) because it represents "dense, global motion" (e.g., full-body or camera movement). In contrast, the real data (blue) is sparser (mean 4.65) and "flat" (mean 0.0179) because it represents "sparse, local motion" (e.g., only hands clapping while the body and background are static).

(b) Artifact Patterns (Sensor vs. Algorithm): The spatial heatmaps (Fig. A1, Plot 3) confirm that each domain has unique, high-intensity artifacts. The real data (Plot 3, Top) exhibits sensor-specific noise (e.g., a horizontal line artifact with a peak intensity of $\sim$60). The synthetic data (Plot 3, Bottom) features algorithmic artifacts (e.g., "blocky" background noise with a peak intensity of $\sim$30) from the conversion process.

**Conclusion (Robustness via Few-Shot):** Our HSFE model must learn from a "difficult" pre-training environment (dense, global motion + algorithmic artifacts). The fact that this pre-trained model performs exceptionally well on the real-world dataset under a few-shot setting (as shown in Sec. 4.3) is the strongest proof of its robustness. It successfully bridges this significant domain gap, demonstrating that it learned the transferable, underlying dynamics of motion itself, rather than overfitting to either domain's specific statistics or noise patterns.

## A.3 ADAPTATION FOR TRADITIONAL MODELS

As a comparative study, we engineered a hybrid visual encoder to investigate whether a specialized spike feature extractor could effectively bridge the modality gap for a conventional Vision Transformer (ViT) backbone. This architecture, encapsulated in the `HybridSpikeEncoder` module, first employs our Hierarchical Spike Feature Extractor (HSFE) as a sophisticated frontend. The HSFE processes overlapping temporal blocks from the raw spike stream, translating the sparse, event-driven

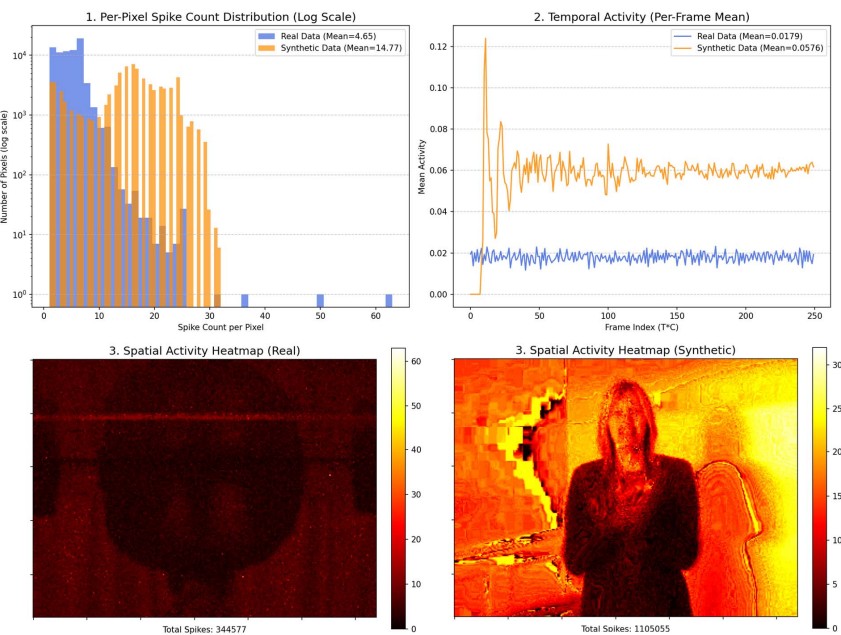

Figure A1: Quantitative and qualitative analysis of the Domain Gap between our synthetic data (SpikeCV) and real-world data. The plots reveal fundamental differences in (1) per-pixel spike distribution (synthetic is denser, mean 14.77 vs. 4.65), (2) temporal activity (synthetic is 'bursty' global motion, real is 'flat' local motion), and (3) spatial artifact patterns (synthetic 'block' noise vs. real 'sensor line' noise).

data into a sequence of dense, feature-rich maps. These maps are then tokenized via a convolutional "bridge" layer and subsequently fed into the ViT.

However, this approach yielded poor performance, failing to converge and achieving less than 40% accuracy. We attribute this to a fundamental mismatch in inductive bias. The ViT is architected to find patterns in the dense, spatially-correlated information of natural images. The feature maps produced by the HSFE, while rich in temporal dynamics, do not possess the pixel-like spatial qualities that the ViT is primed to learn from. This experiment demonstrates that merely adapting the input is insufficient; effective spike video understanding requires a holistically designed architecture rather than a simple fusion of disparate components.

### A.4 Validating Multimodal Alignment via Text-to-Video Retrieval

To validate that our model learns a meaningful joint embedding space between spike videos and language—rather than merely performing single-modal classification—we conducted a challenging text-to-video retrieval experiment.

The experimental setup was designed to be rigorous. Instead of relying on simple, single-category keywords (e.g., "brushing teeth"), we leveraged the unique and semantically rich captions associated with each video in the HMDB51-S dataset as text queries. This instance-level retrieval task places higher demands on the model, requiring it to grasp subtle correspondences between high-speed video content and nuanced natural language descriptions.

Under this challenging setting on the HMDB51-S dataset, our model demonstrated strong performance. It achieved a **Recall@1 (R@1) of 31.94%**, a **R@5 of 63.12%**, and a **R@10 of 75.10%**. These results provide direct, quantitative evidence that our model successfully learns a deep semantic alignment between spike videos and text, confirming the multimodal capabilities central to our framework's design.

### A.5 Implementation Details of the Full-Spiking Visual Encoder (FSVE)

Building on SPKLIP, we propose a FSVE tailored for event streams of spike camera. Through the synergistic design of MS-ResNets (Hu et al., 2025) and Spiking Temporal Transformer, we achieve end-to-end spatiotemporal feature learning in the pure spiking domain. The architecture is illustrated in Fig. 3.

Spiking ResNets extract spatial features with temporal-dependent normalization. To exploit SNNs' inherent compatibility with spike data, we adapt MS-ResNets with spiking dynamics:

(1) Replace continuous activations with LIF neurons:

$$\mathcal{S}^{(t)} = \begin{cases} 1 & \text{if } u^{(t)} \geq \text{thresh} \\ 0 & \text{otherwise} \end{cases} \tag{7}$$

(2) Introduce temporal-dependent Batch Normalization (TDBN) to stabilize membrane potential evolution across time steps; (3) Define spiking residual function: $\mathcal{S}_{l+1} = f_{\text{spike}}\left(\text{TDBN}(\mathcal{F}_{\text{spike}}(\mathcal{S}_l)) + \mathcal{S}_l\right)$ where $f_{\text{spike}}$ converts membrane potentials to binary spikes $\{0, 1\}$, and $\mathcal{F}_{\text{spike}}$ denotes spiking convolution. For backpropagation, we use a rectangular surrogate gradient:

$$\frac{\partial \mathcal{S}}{\partial u} \approx \frac{1}{2\text{lens}}\mathbb{I}\left(|u - \text{thresh}| \leq \text{lens}\right) \tag{8}$$

with lens controlling gradient window width.

Spiking Temporal Transformer enables energy-efficient spatiotemporal correlation learning. We adapt an efficient E-SDSA module (Yao et al., 2025a) and tailor it for spike-based vision tasks. The module integrates two key components (Fig. 3b):

1. Spike-encoded QKV generation with threshold normalization: Query/key/value projections use linear layers followed by spike normalization:

$$Q_S = \text{SN}(\text{Linear}(U)), \quad K_S = \text{SN}(\text{Linear}(U)), \quad V_S = \text{SN}(\text{Linear}(U))$$
$$\text{SN}(x) = \Theta(x - V_{\text{th}}), \quad V_{\text{th}} = \alpha \cdot \mathbb{E}[|x|] \tag{9}$$

where $\Theta$ is the Heaviside function, and $\alpha$ is a learnable scaling factor. This sparse encoding reduces energy consumption compared to analog QKV generation.

2. Sparse self-attention computation with threshold reparameterization: The attention operator computes sparse correlations via:

$$U' = \text{Linear}\left(\text{SN}\left(\frac{Q_S \cdot K_S^{\top}}{\sqrt{d}} \odot \text{scale}\right) \cdot V_S\right) \tag{10}$$

Threshold reparameterization stabilizes learning:

$$V_{\text{th}}' = \frac{V_{\text{th}}}{\text{scale}} \tag{11}$$

This design achieves two advantages: (1) Event-driven sparsity reduces computation; (2) Threshold reparameterization stabilizes attention learning under varying input dynamics.

### A.6 SNN Energy Consumption Analysis

To evaluate the energy efficiency of our full-spiking architecture, we developed a detailed energy model that accounts for both computational operations and crucial memory access costs. Our model utilizes established energy cost parameters from Horowitz's research on 45nm CMOS process technology (Horowitz, 2014).

The parameters adopted for our estimation are as follows:

- **Computational Cost** ($E_C$)**:** 4.6 pJ per 32-bit operation (for both MAC and AC).
- **Neuron Update Cost** ($E_U$)**:** 0.9 pJ per 32-bit operation.
- **Memory Access Cost** ($E_M$)**:** 5.0 pJ per 32-bit read/write from SRAM.

Based on these parameters, the energy models for the standard Artificial Neural Network (ANN) and our Spiking Neural Network (SNN) architectures are formulated.

**ANN Energy Model**  The energy for the dense ANN baseline is the sum of its computational and memory access costs.

$$E_{\text{ANN}} = (\text{Total\_Ops} \times E_C) + (\text{Memory\_Reads} \times E_M) \qquad (12)$$

**SNN Energy Model**  The energy for the sparse SNN model accounts for actual synaptic operations (SOPs), neuron potential updates, and all associated memory accesses.

$$E_{\text{SNN}} = (\text{Actual\_SOPs} \times E_C) + (\text{Neuron\_Updates} \times E_U) + (\text{Memory\_Accesses} \times E_M) \qquad (13)$$

The detailed breakdown of this analysis is presented in Table 6. The results highlight a critical trade-off in SNN efficiency. While the SNN architecture reduces computational energy by an estimated 74.1% due to its inherent data sparsity, it also introduces substantial memory access overhead for updating and retrieving neuron membrane potentials.

Table 6: Detailed energy consumption analysis including memory access costs.

| Key Metrics | ANN (Baseline) | SNN (Our Model) | Analysis |
|---|---|---|---|
| Computational Energy | 1.372 J | 0.356 J | 74.1% reduction due to sparsity |
| Memory Access Energy | 0.0036 J | 0.791 J | Higher due to membrane potential updates |
| **Total Estimated Energy** | **1.375 J** | **1.147 J** | **16.6% overall energy saving** |
| Memory Energy Ratio | 0.26% | 68.99% | Bottleneck shifts to memory access |

This analysis reveals that SNNs often transform a **compute-bound** problem into a **memory-bound** one, where frequent memory access becomes the new energy bottleneck. In our SNN model, memory-related operations account for 68.99% of the total energy. Despite this shift, the substantial reduction in computational requirements leads to a notable net energy saving of 16.6%, demonstrating the overall efficiency advantage of the full-spiking approach.

### A.7 Exploring Full-Spike Dynamics: Architecture and Efficiency of SPKLIP

We also explored the performance of a full-spike dynamics model. To evaluate our framework's energy efficiency, we implemented a full-spiking version by converting the components of the visual encoder to Spiking Neural Networks (SNNs). When the CNN part of the original visual encoder was replaced with its SNN counterpart, the model's accuracy on the UCF101-S dataset decreased to 74.14%. When all components of the visual encoder (including the Transformer) were converted to SNNs, the performance dropped to 67.29%.

Based on an estimation model, the SNN architecture achieves an approximate 74.12% reduction in computational energy compared to the standard ANN baseline, primarily due to the inherent computational sparsity of SNNs.

This efficiency gain is also accompanied by a trade-off in accuracy. A detailed analysis of this architecture, the energy estimation methodology, and results is available in Appendix A.5 and Appendix A.6.

### A.8 Video-to-Spike Preprocessing Pipeline

We design a two-stage preprocessing pipeline to convert conventional video data into standard spike event streams: neural network-based frame interpolation and spike encoding.

#### A.8.1 Frame Interpolation for Enhanced Temporal Resolution

Raw video frames from action recognition datasets (e.g., UCF101 and HMDB51) are processed through a pre-trained video frame interpolation model. The model architecture contains:

- `Feature_extractor`: Extracts hierarchical spatial features
- `MultiScaleFlow.block`: Estimates MultiScale optical flow
- `Unet`: Refines residual details via bidirectional optical flow guidance and mask fusion

The interpolation synthesizes intermediate frames using bidirectional alignment, mask fusion, and residual correction. Temporal expansion factors are applied:

- UCF101: $\times 10$ frame rate expansion
- HMDB51: $\times 50$ frame rate expansion

Output sequences are formatted as 4D tensors $[T, H, W, C]$ where:

- $T$: Temporal dimension
- $H \times W$: Spatial resolution
- $C = 3$: RGB channels

### A.8.2 SPIKE ENCODING VIA TEMPORAL INTEGRATION

High-frame-rate RGB videos are converted to spike data through our encoding algorithm:

1. Frame conversion to grayscale with pixel normalization $[0, 1]$
2. Membrane potential accumulation: $V_t = V_{t-1} + I_t$
3. Spike generation:

$$\text{spike matrix}[t, x, y] = \begin{cases} 1 & \text{if } V_t(x, y) \geq \theta \\ 0 & \text{otherwise} \end{cases}$$

   with threshold $\theta = 5.0$ and potential reset $V_t \leftarrow V_t - \theta$ after spike
4. Repeat until all frames processed

The `stack_to_spike` function generates binary spike tensors $[T, H, W]$ with configurable:

- Additive noise injection
- Threshold $\theta$ adjustment

Final serialization via `SpikeToRaw` function:

- Encodes 8 spikes per byte (binary compression)
- Outputs .dat files for SPKLIP compatibility
- Decoding reconstructs Boolean tensor $[T, H, W]$ during inference

The proposed two-stage preprocessing pipeline effectively bridges conventional videos and neuro-morphic vision processing. By combining deep learning-based frame interpolation with bio-inspired spike encoding, we achieve:

- **Temporal Super-Resolution**: Neural interpolation extends temporal sampling density by 10-50× through multi-scale optical flow and attention mechanisms, preserving physical consistency in dynamic scenes
- **Biologically Plausible Encoding**: The temporal integration algorithm emulates retinal neuron dynamics, converting intensity variations into sparse spike events with adaptive threshold control
- **System Compatibility**: Serialized spike data (.dat) with byte-level compression ensures seamless integration with SPKLIP-based neuromorphic classifiers

This pipeline enables efficient conversion of standard video datasets into spike-compatible formats while maintaining configurable spatiotemporal properties, establishing a practical foundation for spike-based action recognition research.

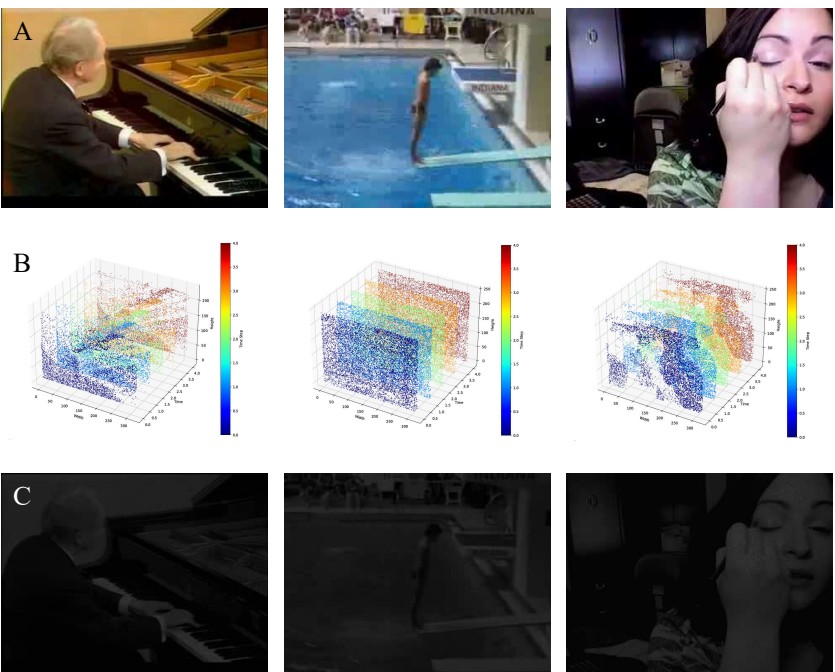

Figure A1: This figure displays three components, A: the first frame of the original RGB video from the UCF101 dataset, B: the spike lattices of the first five timesteps from the converted .dat file, C: the first frame of the reconstructed grayscale video generated through the TFI conversion process.

To validate dataset conversion accuracy, we employed the Texture From Interval (TFI) algorithm from the SpikeCV toolkit to reconstruct grayscale images from $[T, H, W]$-dimensional spike tensors. As it is shown in Fig A1. This algorithm leverages the spatiotemporal sparsity and informational potential of spike signals to approximate the texture structures of conventional images.

The TFI principle posits that temporal intervals between adjacent spikes reflect texture intensity: shorter intervals indicate higher pixel activity and correspondingly brighter intensity. Specifically, TFI calculates the nearest two spike timestamps within a maximum temporal window ($\pm\Delta t$) around each target moment, then derives pixel-wise grayscale values based on their interval duration.

## B    TECHNICAL APPENDICES AND SUPPLEMENTARY MATERIAL

The source code and dataset are available at [link removed for anonymity].

