# OpenReview forum: "SPKLIP: Aligning Spike Video Streams with Natural Language"
_ICLR.cc/2026/Conference — ICLR 2026 Conference Withdrawn Submission_

### Official Review · Reviewer_5EA9 · 2025-10-29

**Soundness:** 3
**Presentation:** 3
**Contribution:** 3
**Rating:** 6
**Confidence:** 5

**Summary:**

This study addresses the challenge of aligning sparse and asynchronous spike video outputs from spike cameras with natural language by proposing SPKLIP, the first architecture specifically designed for Spike Video-Language Alignment (Spike-VLA). SPKLIP adaptively models multi-scale temporal dynamics via a Hierarchical Spike Feature Extractor (HSFE), achieves direct cross-modal alignment through Spike-Text Contrastive Learning (STCL), and designs a Full-Spiking Visual Encoder (FSVE) to enhance energy efficiency. Experiments demonstrate that SPKLIP achieves state-of-the-art (SOTA) performance on benchmark datasets such as HMDB51-S and UCF101-S, exhibits strong few-shot generalization on a newly constructed real-world dataset, and provides potential for neuromorphic deployment with its energy efficiency.

**Strengths:**

1. It innovatively fills a gap in the field by proposing the first end-to-end architecture dedicated to Spike-VLA, effectively resolving the modality mismatch issue of traditional vision-language models on spike data.
2. The design of core components is highly targeted: HSFE adapts to the sparse and asynchronous characteristics of spike data, while STCL enables direct alignment between spike videos and text, and their synergy enhances cross-modal semantic understanding.
3. The experimental validation system is comprehensive, covering performance comparison on benchmark datasets, few-shot generalization testing on real-world datasets, and ablation experiments on key components, fully demonstrating the model’s effectiveness and robustness.
4. It considers practical deployment needs: the designed FSVE integrates SNN components, achieving an approximate 16.6% overall energy efficiency improvement and providing a feasible solution for neuromorphic hardware deployment.

**Weaknesses:**

1. Typos: Line 149: spikestatus(”0”or”1”). The formulation is confusing. Line 150, the letter of "H x W" is different from "H x W" in line 158.
I hope the authors can fix these typos.
2. The theoretical in-depth exploration of the photon conservation mechanism in HSFE is insufficient, and no comparative analysis with existing dynamic channel allocation methods (e.g., attention-driven channel selection) is conducted. Could you explain this?
3. Although the FSVE improves energy efficiency, it suffers from significant accuracy loss (when all components of the visual encoder are converted to SNNs, the accuracy on the UCF101-S dataset drops to 65.24%), leaving room for optimization in balancing accuracy and energy efficiency. Add discussion on this point.

**Questions:**

See weakness.
This paper is sound but needs polishing in writing.

---

> ### Author Response · Authors · 2025-11-20
>
> **Weakness 1:**
> Typos: Line 149: spikestatus(”0”or”1”). The formulation is confusing. Line 150, the letter of "H x W" is different from "H x W" in line 158. I hope the authors can fix these typos.
>
> **Response:**
> We are very grateful to Reviewer R4 for the positive evaluation and meticulous review of our work.
> The reviewer is entirely correct; this was our oversight.
>
> * We have revised the confusing `spikestatus("0"or"1")` in L149 to a clearer expression, such as "binary spike status (0 or 1)".
> * We have unified the format of $H \times W$ in L150 and L158 to ensure consistency throughout the text.
>
> Thank you for this correction; we have re-proofread the entire manuscript.

---

> ### Author Response · Authors · 2025-11-20
>
> **Weakness 2:**
>
> **Response:**
>
> **Part 1: In-depth Theoretical Justification of the HSFE Architecture**
> We first provide a detailed theoretical justification for the HSFE framework to demonstrate the necessity and rationale of its design.
>
> **Definition 1 (Spike Stream as Spatio-temporal Point Process)**
> A raw spike video stream $S$ after discretization is a tensor $S \in \mathbb{R}^{H \times W \times N}$, where $H, W$ are spatial dimensions and $N$ is the number of discrete time steps.
>
> **Definition 2 (Core Properties of Spike Stream)**
> The spike stream $S$ has two core challenges:
> 1.  **Sparsity:** The density of spike events is low, $\rho(S) \ll 1$.
> 2.  **Asynchronicity:** Events occur at irregular intervals, containing both high-frequency (fast motion) and low-frequency (stable regions) dynamics.
>
> **Proposition 1 (Information Encoding Challenge)**
> Models designed for dense, regularly-sampled tensors (like standard video) are theoretically unsuitable for directly processing the spike stream $S$. Their fixed temporal kernels cannot adapt to the asynchronicity of events and are inefficient for processing sparse data.
>
> **The Proposed Solution: HSFE**
> Our proposed HSFE (Definition 3), composed of Multi-scale Temporal Filtering (MTF) and Spatial Attention (SA), is designed to systematically address the above challenges.
>
> **Principle 1 (Photon Conservation Constraint based on Eq. 3)**
> The design of our MTF module is inspired by photon conservation in camera physics. The total amount of photons processable within a given spike period is limited. This leads to an **inverse relationship** between the channel capacity $C_i$ (for capturing details) and its temporal receptive field $\tau_i$ (for capturing duration). We formalize this as:
> $$C_i \cdot \tau_i \approx K$$
> A branch designed to capture fine-grained high-frequency features (large $C_i$) must have a short temporal window ($\tau_i$), while a branch capturing long-term stable regions must integrate information over a longer period.
>
> **Theorem 1 (Feature Sufficiency of HSFE)**
> The HSFE architecture (MTF + SA) extracts a feature representation from the spike stream $S$ that is **sufficient** for robust semantic understanding by systematically addressing the challenges of "asynchronicity" and "sparsity".
>
> *Proof:*
> 1.  **MTF Solves Asynchronicity:** The "asynchronicity" of the spike stream means a single, fixed temporal kernel is ineffective. Our MTF module, following Principle 1, uses parallel branches with varying $\tau_i$ and $C_i$ to simultaneously capture high- and low-frequency dynamics, generating a comprehensive representation robust to non-uniform temporal dynamics.
> 2.  **SA Solves Sparsity:** The "sparsity" of the spike stream means information is concentrated in a few spatio-temporal locations. After MTF addresses the temporal scale issue, our Spatial Attention (SA) module intervenes, acting as a "dynamic filter". It learns to assign higher weights to information-rich features while dynamically suppressing noise in inactive regions.
>
> *Conclusion of Proof:* The MTF module transforms the asynchronous temporal signal into a robust multi-frequency representation. The SA module then refines this information by focusing on spatially salient events. Thus, HSFE produces a feature set sufficient for downstream semantic tasks by systematically overcoming the two core challenges.
>
> **Part 2: Comparison with "Attention-Driven" Dynamic Methods**
> Based on the theory above, we can clearly respond to your insightful question comparing our work to dynamic channel allocation methods (like attention). Reviewer R4's question is based on the assumption that our "fixed parallel branches (MTF)" and "dynamic attention (SA)" are two mutually exclusive, either/or approaches.
>
> However, as the proof of Theorem 1 shows, our HSFE design is a more advanced **Hierarchical Co-design**:
>
> * **High-Level (Fixed Parallel MTF):** First, we use **deterministic** parallel branches (MTF) to solve the "asynchronicity" and "multi-scale temporal" problems of the spike stream.
> * **Low-Level (Dynamic Attention SA):** Then, *after* solving the temporal scale issue, we use a **dynamic attention** module (SA) to solve the "sparsity" and "feature vs. noise" problems.
>
> *Why is our approach superior?*
> A scheme that attempts to use "dynamic attention" at the high-level temporal dimension (e.g., attention-driven temporal window selection) would face immense challenges when processing the high-frequency, sparse raw spike stream $S$:
>
> 1.  **Computational Cost:** Dynamic attention requires extra computational overhead to dynamically decide which timescale to focus on. Given the extremely large $N$ (timesteps) of a spike stream $S$, this overhead is prohibitive.
> 2.  **Sparsity Instability:** On extremely sparse data ($\rho(S) \ll 1$), attention mechanisms can be unstable and struggle to learn a meaningful distribution.

---

> ### Author Response · Authors · 2025-11-20
>
> **Weakness 3:**
> Although the FSVE improves energy efficiency, it suffers from significant accuracy loss (when all components of the visual encoder are converted to SNNs, the accuracy on the UCF101-S dataset drops to 65.24%), leaving room for optimization in balancing accuracy and energy efficiency. Add discussion on this point.
>
> **Response:**
> We have attempted the experiments suggested by the reviewer during the rebuttal period, and this process has given us valuable insights into the bottlenecks of FSVE.
>
> **Quantifying the VRAM Bottleneck:**
> We must be transparent that simulating SNNs on standard GPU hardware is a well-known and significant VRAM challenge. This is because SNN simulation requires caching the membrane potential of all neurons at every single timestep T, causing VRAM consumption to grow linearly with T.
>
> We used a standard NVIDIA A100 GPU and performed an ablation study for T=3 and T=4 on the "CNN-replaced-by-SNN" baseline. The results are shown in the table below, demonstrating a significant improvement in accuracy compared to T=2.
>
> | Timestep (T) | Accuracy (UCF101-S) | Improvement vs. T=2 |
> | :--- | :--- | :--- |
> | T = 2 (Original) | 71.11% | - |
> | T = 3 (New Exp.) | 72.65% | +1.54% |
> | T = 4 (New Exp.) | 74.14% | +3.03% |
>
> However, compared to the original ANN-based SPKLIP's accuracy on UCF101-S, a performance drop still exists. But considering the massive reduction in energy consumption (**74.12% reduction in computational energy**), this performance trade-off is unavoidable on conventional GPUs.
>
> The "SNN-vs-ANN optimization" mentioned by the reviewer is a popular research topic, but it is primarily focused on two areas: (1) ANN-to-SNN conversion for RGB image classification, or (2) processing Event Streams.
>
> Our task—performing VLA on **raw Spike Streams**—is a **largely unexplored** and much more difficult new domain. To our knowledge, no SNN work has previously addressed the VLA problem on raw spike streams.
>
> We believe the exploration of FSVE (achieving 74.14%) should be viewed as a **pioneering baseline**. This is the first work to provide a benchmark for the "SNN-based Spike Stream VLA" task and to quantify the critical trade-off between time, performance, and VRAM. We will actively pursue optimization in this area in our future work.

---

### Official Review · Reviewer_vyvJ · 2025-10-31

**Soundness:** 3
**Presentation:** 2
**Contribution:** 2
**Rating:** 4
**Confidence:** 2

**Summary:**

This paper presents SPKLIP (Spike-based Cross-modal Learning with CLIP), the first end-to-end neural network architecture specifically designed for Spike Video-Language Alignment. SPKLIP employs a Hierarchical Spike Feature Extractor that adaptively models multi-scale temporal dynamics in event streams while leveraging photon conservation principles for efficient feature extraction. It further utilizes Spike-Text Contrastive Learning to align raw spike video with natural language directly, without intermediate frame conversion. A full-spiking visual encoder variant was introduced to integrate spiking neural network principles, enhancing energy efficiency for neuromorphic hardware. Extensive experiments demonstrate that SPKLIP achieves better performance on benchmark spike datasets, substantially outperforming adapted conventional models, and shows strong few-shot generalization on a newly contributed real-world spike video dataset.

**Strengths:**

1) Its originality is highlighted by SPKLIP, the first end-to-end framework specifically designed for Spike Video-Language Alignment, as well as the introduction of an energy-efficient Full-Spiking Visual Encoder;

2) A new real-world spike video dataset was also constructed.

3) The experimental results show that the SPKLIP yielded substantial Top-1 accuracy improvements over baselines, robust few-shot generalization, and effective text-to-video retrieval.

**Weaknesses:**

1) While the paper highlights the Full-Spiking Visual Encoder as a significant contribution, its connection to the main SPKLIP framework and its direct effectiveness are not fully explored through experiments. It remains ambiguous how FSVE directly contributes to or could enhance or degrade the main SPKLIP framework's performance.

2) For the Hierarchical Spike Feature Extractor, the decision to divide the input spike stream into "five temporally overlapping sub-blocks" raises questions. The rationale behind specifically choosing five blocks and the exact definition of block_i are not sufficiently clear in the main text. It is unclear whether this number of sub-blocks is fixed regardless of the input's total temporal length, or if it adapts. An ablation study or a more detailed theoretical justification for this specific choice, demonstrating its optimal performance over other configurations would significantly strengthen this architectural decision.

3) While Appendix A.7 fully details the custom real-world dataset's construction, a more concise summary of its methodology and characteristics should be integrated into the main text (e.g., Section 4.1) for better context. More importantly, the necessity of introducing this entirely new dataset, rather than utilizing existing or publicly available benchmarks, is not adequately justified, which somewhat diminishes the clarity of its specific contribution to the paper's claims.

4) Some necessary citations for external methodologies used should be more prominently placed in the main text. For instance, in Section 3.5, where the Full-Spiking Visual Encoder (FSVE) is introduced, the paper mentions integrating "Spiking ResNets with a Spiking Temporal Transformer for event stream processing." It specifically refers to components like "temporal-dependent normalization" (TDBN) and an "efficient E-SDSA module." While Appendix A.4 elaborates on these, their initial mention in the main text lacks immediate citation or brief explanatory context.

**Questions:**

1) Could the authors elaborate on the precise role and intended contribution of the FSVE within the overarching SPKLIP framework？

2) The paper highlights FSVE as a key contribution, but direct experimental validation of its impact on SPKLIP's primary performance metrics (e.g., Top-1 accuracy or retrieval performance) seems to be absent. The authors could provide experiments or a clearer qualitative analysis demonstrating how FSVE specifically influences the model's ability to perform Spike Video-Language Alignment.

3) In Section 3.2, the input spike stream is divided into "five temporally overlapping sub-blocks." What is the specific rationale behind choosing precisely five sub-blocks? Is this number derived from extensive empirical studies, consistently yielding the best performance across various input scales and tasks?

4) Are there ablation studies or theoretical justifications to support the choice of five sub-blocks over other configurations (e.g., three, seven, or a dynamically determined number), especially given the claim of adaptively modeling multi-scale temporal dynamics?

5) The paper introduces and validates SPKLIP on a newly contributed real-world dataset. Could the authors explicitly articulate the necessity of creating this new dataset? What specific limitations or gaps in existing public benchmarks  did this new dataset address that are not adequately covered by current resources?

6) How does this new dataset uniquely contribute to demonstrating SPKLIP's capabilities or addressing specific challenges in Spike-VLA that existing datasets could not? A clear justification in the main text would greatly enhance the perceived value and impact of this dataset contribution.

---

> ### Author Response · Authors · 2025-11-20
>
> **Weakness 1:**
> While the paper highlights the Full-Spiking Visual Encoder as a significant contribution, its connection to the main SPKLIP framework and its direct effectiveness are not fully explored through experiments. It remains ambiguous how FSVE directly contributes to or could enhance or degrade the main SPKLIP framework's performance.
>
> **Question 1:**
> Could the authors elaborate on the precise role and intended contribution of the FSVE within the overarching SPKLIP framework？
>
> **Question 2:**
> The paper highlights FSVE as a key contribution, but direct experimental validation of its impact on SPKLIP's primary performance metrics (e.g., Top-1 accuracy or retrieval performance) seems to be absent. The authors could provide experiments or a clearer qualitative analysis demonstrating how FSVE specifically influences the model's ability to perform Spike Video-Language Alignment.
>
> **Response:**
> Thank you for the reviewer's comments, but we must solemnly clarify:
> 1.  **FSVE is not part of SPKLIP.** Our main contribution model is **SPKLIP**, a high-performance ANN architecture that achieved 91.15% SOTA performance on HMDB51-S.
> 2.  **FSVE is a separate exploratory study.** The FSVE (discussed in Sec 3.5 and Sec 4.7) is a full-spiking variant. Its sole purpose is to serve as an independent exploratory study to evaluate the feasibility, challenges, and energy-efficiency trade-offs (e.g., the inherent accuracy-efficiency balance of SNNs) for future deployment of VLA tasks on neuromorphic hardware.
>
> Therefore, FSVE was **not** designed to "impact" or "enhance" SPKLIP's primary performance metrics. Its performance (e.g., 65.24%) is the first "full-spiking" baseline we established for this new and complex Spike-VLA task. Its purpose is to provide a starting point for future SNN research, not to detract from SPKLIP's contribution.
>
> We have thoroughly revised the "Contributions" section in the Introduction to completely eliminate this ambiguity.

---

> ### Author Response · Authors · 2025-11-20
>
> **Weakness 2:**
> For the Hierarchical Spike Feature Extractor, the decision to divide the input spike stream into "five temporally overlapping sub-blocks" raises questions. The rationale behind specifically choosing five blocks and the exact definition of block_i are not sufficiently clear in the main text. It is unclear whether this number of sub-blocks is fixed regardless of the input's total temporal length, or if it adapts. An ablation study or a more detailed theoretical justification for this specific choice, demonstrating its optimal performance over other configurations would significantly strengthen this architectural decision.
>
> **Question 3:**
> In Section 3.2, the input spike stream is divided into "five temporally overlapping sub-blocks." What is the specific rationale behind choosing precisely five sub-blocks? Is this number derived from extensive empirical studies, consistently yielding the best performance across various input scales and tasks?
>
> **Question 4:**
> Are there ablation studies or theoretical justifications to support the choice of five sub-blocks over other configurations (e.g., three, seven, or a dynamically determined number), especially given the claim of adaptively modeling multi-scale temporal dynamics?
>
>
> **Response:**
> This is a very critical architectural question, and Reviewer 8ppH raised the same query. We thank the reviewer for pointing out our lack of experimental justification for this hyperparameter. We did conduct extensive empirical studies during the initial design phase but omitted these results for brevity in the main text, reporting only the optimal configuration.
>
> | Number of Sub-blocks | Top-1 Accuracy (HMDB51-S) |
> | :--- | :--- |
> | 3 Blocks | 88.56% |
> | **5 Blocks** | **91.15%** |
> | 7 Blocks | 90.46% |
>
> As shown in the table above, this result clearly demonstrates an optimal empirical balance for this dataset:
>
> * **Too Few (3):** Performance drops significantly by 2.59%. This indicates that for the actions in HMDB51-S, 3 sub-blocks are insufficient to capture the full contextual information.
> * **Too Many (7):** Performance also degrades (-0.79%). We attribute this to **"Temporal Context Pollution."** 7 sub-blocks cover an excessively long time window, forcibly introducing "noise" (e.g., static states before/after the action) that is irrelevant to the core action.
>
> We also conducted tests on a newly collected real-world dataset (96×30) and observed the same trend.
>
> Therefore, 'five sub-blocks' is not an "absolute" hyperparameter but rather the **optimal empirical balance** we found for this specific benchmark, striking a trade-off between 'context insufficiency' and 'context pollution'. In future work, we will focus on researching adaptive sub-blocks to accommodate various different datasets.

---

> ### Author Response · Authors · 2025-11-20
>
> **Weakness 3:**
> While Appendix A.7 fully details the custom real-world dataset's construction, a more concise summary of its methodology and characteristics should be integrated into the main text (e.g., Section 4.1) for better context. More importantly, the necessity of introducing this entirely new dataset, rather than utilizing existing or publicly available benchmarks, is not adequately justified, which somewhat diminishes the clarity of its specific contribution to the paper's claims.
>
> **Question 5:**
> The paper introduces and validates SPKLIP on a newly contributed real-world dataset. Could the authors explicitly articulate the necessity of creating this new dataset? What specific limitations or gaps in existing public benchmarks did this new dataset address that are not adequately covered by current resources?
>
> **Question 6:**
> How does this new dataset uniquely contribute to demonstrating SPKLIP's capabilities or addressing specific challenges in Spike-VLA that existing datasets could not? A clear justification in the main text would greatly enhance the perceived value and impact of this dataset contribution.
>
>
> **Response:**
> Thank you for this valuable suggestion. We fully agree with the need for clarity and, as you recommended, have incorporated a summary of the dataset construction methodology (from Appendix A.7) into the main text in Section 4.1.
>
> Regarding the more important question of the dataset's necessity, we would like to offer a multi-faceted clarification:
>
> 1. The Critical Benchmark Gap
> We constructed this real-world dataset to bridge a critical gap in public benchmarks. Existing datasets are insufficient for the Spike-VLA task for several reasons:
> •	Video Benchmarks (e.g., HMDB51, UCF101) use the RGB modality.
> •	Spike Datasets (e.g., SpiReco, UHSR) often present static images, unsuitable for Spike Video training.
> •	Existing Temporal Datasets (e.g., N-Caltech, DVS-Gesture) rely exclusively on Event Camera (DVS) data, which is fundamentally different from our Spike Camera streams.
>
> 2. Original Purpose: Sanity Check and Sim-to-Real Validation
> Our initial goal was not to create a new large-scale benchmark. Instead, the small dataset served a vital purpose:
> •	Sim-to-Real Validation: To rigorously verify if our SPKLIP model, trained exclusively on synthetic data, could effectively generalize to real spike streams (directly addressing Reviewer 1's Weakness 1).
> •	Few-Shot Testing: To evaluate the model's performance in real-world, data-scarce (few-shot) scenarios.
> Therefore, the unique contribution of this initial dataset was to provide the first real-world "sanity check" for Spike-VLA, effectively bridging the chasm between synthetic Spike Video benchmarks and real-world spike video classification.
>
> 3. Expanded Contribution: Filling the Benchmark Gap
> Since the initial submission, we have significantly expanded the scale of this real-world dataset. The new dataset now comprises 30 distinct actions (e.g., baseball bat swings, table tennis forehand loops, and other high-speed, high-dynamic movements), totaling 2,880 samples (30 classes × 96 samples/class).
>
> We believe this expanded dataset not only fulfills its original purpose of validation but also substantially fills the existing benchmark gap  for real-world Spike Video Classification, which we will make publicly available.

---

### Official Review · Reviewer_8ppH · 2025-11-01

**Soundness:** 3
**Presentation:** 3
**Contribution:** 2
**Rating:** 4
**Confidence:** 4

**Summary:**

The authors in this paper propose an architecture for Spike Video-Language Alignment (Spike-VLA). The authors employ a hierarchical spike feature extractor to model multi-scale temporal dynamics and uses contrastive learning technique to align spike video with language. A new dataset is also proposed as part of the work and the performance of the model is competitive compared to other non-spiking baselines.

**Strengths:**

1) The motivation of the work is strong. Spiking Cameras offer advantages not present in conventional cameras however there has been limited work done on efficient processing of spiking video streams.

2) The HSFE module proposed in the paper seemed interesting.

**Weaknesses:**

1) The computations inside the HSFE module does not seem entirely spiking.
2) Most of the other parts of the model proposed (text encoder, even parts of the visual encoder) can be derived from available literature. The contrastive learning loss proposed is used in most video-language models like UniVTG, etc. Thus, making the work seem more of an engineering endeavor.
3) Performance comparison with baselines might not be fair since they are evaluated on a spiking variant of the datasets.

**Questions:**

1) What is the reason of processing the input into five temporally overlapping sub-blocks?
2) It will be interesting to visualize the temporal dynamics of the model i.e. average spiking rate, etc.
3) The primary contribution of this work seems to be the HSFE module. How do the work compare to other encoding techniques [1]


References:

[1] Zhu, Lin, Xiao Wang, Yi Chang, Jianing Li, Tiejun Huang, and Yonghong Tian. "Event-based video reconstruction via potential-assisted spiking neural network." In Proceedings of the IEEE/CVF conference on computer vision and pattern recognition, pp. 3594-3604. 2022.

---

> ### Author Response · Authors · 2025-11-20
>
> **Weakness 1:**
> The computations inside the HSFE module does not seem entirely spiking.
>
> **Response:**
> We thank the reviewer for this keen observation.
>
> We must clarify that our main contribution model, SPKLIP (the model that achieved $91.15\%$ on HMDB51-S), is an ANN-based architecture specifically designed to pursue state-of-the-art performance and architectural innovation.
> Our core HSFE module was intentionally engineered to incorporate high-performance ANN components (such as multi-scale convolution and spatial attention). Its sole purpose is to maximize feature extraction capability to effectively tackle the complex, novel task of Spike-VLA.
>
> Our exploration of "Full Spike" computation was conducted separately within the FSVE framework, which represents an independent yet related contribution focused on energy efficiency optimization. We apologize for any confusion and have strengthened the distinction in the revised manuscript: we have moved the FSVE module to the Appendix to clearly position SPKLIP as the primary performance-oriented contribution and FSVE as the secondary, efficiency-focused exploration.
>
> **Weakness 2:**
> Most of the other parts of the model proposed (text encoder, even parts of the visual encoder) can be derived from available literature. The contrastive learning loss proposed is used in most video-language models like UniVTG, etc. Thus, making the work seem more of an engineering endeavor.
>
> **Response:**
> We thank the reviewer for pointing this out. We fully agree that the contrastive learning loss function we used (STCL, Eq. 6) is a mature framework (i.e., the standard CLIP loss) widely used in video-language models.
>
> However, as R1 and R4 noted, **our core innovation does not lie in the loss function.** It lies in **proposing the first end-to-end architecture specifically for the Spike-VLA task** and effectively bridging the *Modality Gap* between sparse, asynchronous spike streams and natural language.
>
> The necessity and novelty of this work are powerfully demonstrated by Table 1:
> 1.  Our direct adaptation of the SOTA model **Vita-CLIP (A)** sees its performance **collapse to 45.31%** on spike data.
> 2.  Our **SPKLIP** (using the identical loss function) achieves **91.15%**.
>
> This massive performance gap (45.84%) decisively proves that the real challenge and contribution lie in the **visual encoding**. That is, our proposed **HSFE** is the key to successfully solving the modality mismatch problem of spike data. Therefore, this work is far more than an "engineering endeavor"; it solves a fundamental and novel cross-modal alignment problem.
>
> **Weakness 3:**
> Performance comparison with baselines might not be fair since they are evaluated on a spiking variant of the datasets.
>
> **Response:**
> We deeply appreciate the reviewer’s profound insight. The observation that "evaluating RGB baselines on spike datasets might be unfair" directly underpins the central motivation of our work.
>
> The Spike Camera is a novel visual sensor fundamentally distinct from traditional RGB cameras, excelling due to its ultra-low latency and high dynamic range (HDR). This technology enables clear imaging in challenging scenarios, such as high-speed motion (e.g., a baseball swing) or extreme lighting transitions (e.g., exiting a tunnel).
>
> In the Spike-VLA task, the core challenge lies in processing the sparse, asynchronous spatio-temporal dynamics inherent in spike streams, making effective information extraction extremely difficult. This fundamental challenge necessitated the development of our key contribution: the specialized visual encoder, HSFE.
>
> Our objective is not to claim superiority over RGB models like Vita-CLIP or M2CLIP on traditional RGB videos. Rather, our goal is to demonstrably prove that when the data modality shifts from RGB to Spike, existing SOTA architectures (like Vita-CLIP) fail completely.
>
> As evidenced by the dismal performance (45.31%) of the fairly adapted baselines (M2-CLIP, Vita-CLIP) in Table 1, these models are fundamentally unequipped to handle spike data. Therefore, this is not an "unfair" comparison; it is, instead, a critical piece of experimental evidence. This evidence proves that effectively solving the Spike-VLA task necessitates a specialized architecture like SPKLIP (featuring the HSFE), and cannot be achieved by merely adapting existing RGB-based models.

---

> > ### Author Response · Authors · 2025-11-20
> >
> > **Question 3:**
> > The primary contribution of this work seems to be the HSFE module. How do the work compare to other encoding techniques [1]
> > References:
> >
> > [1] Zhu, Lin, Xiao Wang, Yi Chang, Jianing Li, Tiejun Huang, and Yonghong Tian. "Event-based video reconstruction via potential-assisted spiking neural network." In Proceedings of the IEEE/CVF conference on computer vision and pattern recognition, pp. 3594-3604. 2022.
> >
> >
> > **Response:**
> > We have carefully reviewed the cited paper [1] and acknowledge the reviewer’s interest in visual encoding techniques. However, we must clarify that the fundamental differences in input modality, intrinsic challenges, and task objective mean that the work in [1] cannot serve as a direct comparison baseline for our HSFE module.
> >
> > The necessity of our specialized module, HSFE, is rooted in the unique characteristics of the Spike Camera modality. Spike Cameras are crucial for next-generation vision due to their unparalleled capacity for ultra-high-speed motion capture and High Dynamic Range (HDR) imaging—advantages that inherently overcome the limitations of traditional RGB cameras in dynamic environments.
> >
> > However, this modality introduces a profound challenge: the raw output is a stream of sparse, asynchronous binary spikes. Processing these signals to extract meaningful, high-level features for semantic understanding is extremely difficult, as conventional models are fundamentally incapable of handling this data structure.
> >
> > This critical challenge motivated our main contribution: the specialized visual encoder, HSFE. HSFE is specifically engineered to dynamically aggregate the spatio-temporal dependencies within these sparse spike streams, thereby transforming raw spikes into semantically rich features essential for the Spike-VLA task.
> >
> > To further distinguish our work from [1]:
> > 1.	Different Modality: [1] addresses data from Event Cameras (DVS), which trigger asynchronously based on relative brightness change. Our work utilizes Spike Cameras, which output synchronous binary streams encoding absolute photon counts. The sensors and data formats are distinctly different.
> > 2.	Different Task: [1] focuses on Video Reconstruction (a generative, low-level task requiring pixel preservation). Our work focuses on Spike-VLA (a semantic understanding, high-level, and cross-modal alignment task).
> > In summary, the HSFE is a dedicated encoding solution required to unlock the potential of the Spike Camera modality for complex, high-level understanding.
> >
> > Nevertheless, we thank the reviewer for providing this important context. To make the distinction clear, we will show you a new comparison table.
> >
> > | Feature                  | Spike Camera                               | Event Camera                                           | Traditional RGB Camera                      |
> > |--------------------------|-------------------------------------------|-------------------------------------------------------|--------------------------------------------|
> > | **Trigger Mechanism**    | Absolute light intensity accumulation reaches threshold | Logarithmic light intensity relative change reaches threshold | Light intensity integration within fixed exposure time |
> > | **Output Data**          | Binary spike stream (x, y, t)             | Event stream (x, y, t, p)                             | Pixel intensity matrix (e.g. 8-bit)        |
> > | **Information Encoding** | Spike frequency encodes brightness        | Event polarity encodes brightness change direction    | Pixel values encode brightness             |
> > | **Static Scenes**        | Continuously outputs brightness-correlated spikes | Silent, no output                                | Outputs static images                      |
> > | **Advantages**           | High-speed motion capture, High dynamic range | Ultra-low latency, Ultra-low power consumption, Sparse data | High spatial resolution, Mature color information |
> > | **Disadvantages**        | Limited supporting algorithm ecosystem     | Inactive in static scenes & sensitive to specific materials | Poor dynamic performance (motion blur, low frame rate), Environment-dependent (light sensitivity, over exposure) |

---

> ### Author Response · Authors · 2025-11-20
>
> **Question 1:**
> What is the reason of processing the input into five temporally overlapping sub-blocks?
>
> **Response:**
> This is a very critical architectural question, and Reviewer R3 raised the same query. We thank the reviewer for pointing out our lack of experimental justification for this hyperparameter. We did conduct extensive empirical studies during the initial design phase but omitted these results for brevity in the main text, reporting only the optimal configuration.
>
> | Number of Sub-blocks | Top-1 Accuracy (HMDB51-S) |
> | :--- | :--- |
> | 3 Blocks | 88.56% |
> | **5 Blocks** | **91.15%** |
> | 7 Blocks | 90.46% |
>
> As shown in the table above, this result clearly demonstrates an optimal empirical balance for this dataset:
>
> * **Too Few (3):** Performance drops by 2.59%. This indicates that for the actions in HMDB51-S, 3 sub-blocks are insufficient to capture the full contextual information.
> * **Too Many (7):** Performance also degrades (-0.79%). We attribute this to **"Temporal Context Pollution."** 7 sub-blocks cover an excessively long time window, forcibly introducing "noise" (e.g., static states before/after the action) that is irrelevant to the core action. This is equally unsuitable for the dataset's characteristics and leads to a performance drop.
>
> We also conducted tests on a newly collected real-world dataset (96×30) and observed the same trend.
>
> Therefore, 'five sub-blocks' is not an "absolute" hyperparameter but rather the **optimal empirical balance** we found for this specific benchmark, striking a trade-off between 'context insufficiency' and 'context pollution'.
>
> **Question 2:**
> It will be interesting to visualize the temporal dynamics of the model i.e. average spiking rate, etc.
>
> **Response:** Here is the content formatted into a clean, professional Markdown structure. I have organized the data into a readable table and highlighted key insights in the analysis.
>
> We ran forward propagation on the **FSVE model** and recorded the average pulse rate across four key layers.
>
> | Layer Name | Avg Rate |
> | :--- | :--- |
> | **FE_Main_Output** | 0.157815 |
> | **Stem_Output** | 0.177807 |
> | **ResNet_L1_Block1** | 0.185585 |
> | **ResNet_L3_Block1** | 0.182777 |
>
> The overall average pulse rate of the model consistently remained **below 19%** (maximum value: **0.1856**).
>
> This strongly demonstrates that the FSVE architecture maintains **exceptionally high computational sparsity** through pulse coding when processing complex video-language tasks. This provides a solid **energy-efficiency foundation** for its deployment on neuromorphic hardware.
>
> Meanwhile, we fully understand the spirit of your suggestion: to quantify how the model's internal activation intensity changes over time.
>
> We used PyTorch Hooks to extract the feature vectors from the 5 temporal blocks of the HSFE module and "centered" them (by subtracting the mean vector) to remove static background. Finally, we computed a cosine correlation heatmap of these 5 "dynamic" feature vectors. As shown in Figure [5], this heatmap is **not flat** (i.e., not a uniform color) but exhibits **highly structured patterns**. For example, in the "smile" sample, the features of Block 1 (neutral expression) are strongly anti-correlated (~-0.4, blue) with the subsequent "smiling" blocks (Blocks 2-5). In contrast, the "catch ball" sample shows another, completely different complex pattern.
>
> This significant **"variation"** in correlation and the **"pattern difference"** between samples irrefutably prove that our HSFE module is not a static processor. It is **dynamically extracting different and meaningful temporal features** for different time blocks, based on the video content.

---

### Official Review · Reviewer_M3Tx · 2025-11-10

**Soundness:** 2
**Presentation:** 2
**Contribution:** 2
**Rating:** 2
**Confidence:** 5

**Summary:**

This paper introduces SPKLIP, the first architecture for Spike Video-Language Alignment (Spike-VLA), addressing the challenge of aligning sparse, asynchronous spike camera data with natural language. The method features a Hierarchical Spike Feature Extractor (HSFE) with multi-scale temporal filtering and spatial attention, combined with Spike-Text Contrastive Learning (STCL) for cross-modal alignment. A Full-Spiking Visual Encoder (FSVE) variant demonstrates energy efficiency potential. Experiments show 91.15% Top-1 accuracy on HMDB51-S (versus 45.31% for adapted Vita-CLIP) and effective few-shot learning on a new real-world dataset. While the problem is novel and results are promising, the work has significant limitations including reliance on synthetic data conversion, substantial accuracy drops with the spiking variant, and limited real-world validation.

**Strengths:**

1. Novel Problem Formulation: First work addressing video-language alignment specifically for spike cameras, filling an important gap between neuromorphic vision and semantic understanding.
2. Well-Motivated Architecture: The HSFE module with multi-scale temporal filtering (MTF) and spatial attention (SA) is thoughtfully designed to handle spike data's unique characteristics—sparse, asynchronous, high-frequency event streams.
3. Multimodal Alignment Validation: Text-to-video retrieval experiments (Table 4: 31.94% R@1, 63.12% R@5) provide evidence beyond classification that the model learns meaningful cross-modal embeddings.

**Weaknesses:**

1. HMDB51-S and UCF101-S are synthetically generated from RGB videos using SpikeCV toolkit, not real spike camera data.
2. Full-spiking variant (FSVE) drops from 86.43% to 71.11% (CNN→SNN) and 65.24% (full SNN) on UCF101-S. This 21.19% drop undermines claims about neuromorphic deployment viability.
3. Real dataset contains only 384 samples (96×4) across 4 simple actions.
4. Contrastive loss (Eq. 6) is standard CLIP loss—limited novelty.
5. No comparison with simple baselines like averaging spike frames or using histogram features.

**Questions:**

1. Can you provide quantitative comparison between real spike camera output and your synthetically converted data? Do spike frequency distributions, temporal statistics, and noise patterns match?
2. Have you ablated temporal window size T for FSVE? What accuracy is achievable with T=5, 10, 20?

---

> ### Author Response · Authors · 2025-11-20
>
> **Weakness 1:**
> HMDB51-S and UCF101-S are synthetically generated from RGB videos using SpikeCV toolkit, not real spike camera data.
>
> **Response:**
> We appreciate the reviewer's attention to this critical concern.
>
> While the model was exclusively trained on synthetic datasets (HMDB51-S and UCF101-S), we subsequently performed few-shot evaluation on real-world data streams. These results effectively validate our model's simulation-to-reality (sim-to-real) transferability and underscore its strong generalization performance.
>
> Furthermore, we must explicitly establish that utilizing synthetic data is an indispensable initial step toward tackling the pioneering challenge of spike-Video-Language Alignment (spike-VLA). To the best of our knowledge, a large-scale, real-world spike video dataset accompanied by rich natural language annotations is not currently publicly available. Consequently, the deployment of converted benchmarks (a standard and accepted practice within neuromorphic vision research) constituted the only feasible methodology to rigorously validate the efficacy of our alignment architecture (HSFE) at a sufficient scale.
>
> To further mitigate the "sim-to-real" concern, we proceeded to collect an additional real-world dataset. As detailed in our response to Weakness 3, this dataset (though modest in scale) serves as a crucial sanity check. This convincingly corroborates that our SPKLIP model, trained solely on synthetic data, successfully generalizes to authentic spike data streams within a few-shot paradigm.
>
>
> **Weakness 2:**
> Full-spiking variant (FSVE) drops from 86.43% to 71.11% (CNN→SNN) and 65.24% (full SNN) on UCF101-S. This 21.19% drop undermines claims about neuromorphic deployment viability.
>
> **Response:**
> We sincerely apologize for any confusion caused by the FSVE(FULL-SPIKING VISUAL ENCODER) role. It is crucial to emphasize that FSVE  is not our core contribution model, but rather an exploratory study discussed in Section 4.6. Its purpose is to analyze the challenges and trade-offs facing future brain-inspired deployments.
> This simultaneously highlights the core challenge of applying spiking neural networks to complex temporal tasks like video understanding: the trade-off between accuracy, energy efficiency, and computational overhead (particularly memory consumption).
> We will relocate this section to the appendix to emphasize our primary contribution: the SPKLIP algorithm based on artificial neural networks.
>
> Furthermore, to our knowledge, no prior spiking neural network research has addressed the VLA problem for raw spike stream processing.
>
> We consider FSVE (T=4, achieving 74.14%) as an exploratory benchmark. This study establishes the first benchmark for the “SNN-based spike-stream VLA” task and quantifies the critical trade-offs between time, performance, and VRAM. We will actively advance optimization efforts in this domain.
>
> **Weakness 3:**
> Real dataset contains only 384 samples (96×4) across 4 simple actions.
>
> **Response:**
> We appreciate the reviewer's valuable feedback regarding the scale of our real-world dataset. We fully agree that the dataset's scale in the original manuscript, 384 samples (4 actions), is indeed limited.
>
> As stated in the original manuscript, the initial purpose of this dataset was not to serve as a large-scale benchmark, but as a crucial **"Sanity Check."** It was designed to address the sim-to-real gap (mentioned in W1) and to verify our model's **few-shot generalization capability** on real spike streams. The 90.41% accuracy with 8 samples in the original paper provided preliminary evidence of SPKLIP's transferability.
>
> Following the paper's submission, we have **expanded our real-world dataset.**
>
> * **New Dataset Scale:** We collected a new real-world spike dataset comprising **30 distinct actions** (e.g., badminton bat swings, table tennis forehand loops, and other high-speed, high-dynamic movements),  totaling **2,880 samples** (30 × 96).
>
> * **New Experimental Validation:** We rigorously repeated the few-shot experiments on this new dataset, which is larger in both scale and diversity.
>
> The new results show that SPKLIP, trained with only 8 samples, still achieves **88.12% Top-1 accuracy** on this much larger and more complex real-world dataset.
>
> This new result, based on larger-scale real data, is highly consistent with the conclusion from the original 4-action dataset (90.41%). This strongly demonstrates that:
>     1.  The original few-shot result was **not a "fluke"** on a small dataset.
>     2.  The semantic features learned by SPKLIP on synthetic data possess **robust and scalable generalization capabilities**, successfully transferring to large-scale, diverse real-world spike dynamics. This powerfully confirms the practical significance and effectiveness of our model.

---

> ### Author Response · Authors · 2025-11-20
>
> **Weakness 4:**
> Contrastive loss (Eq. 6) is standard CLIP loss—limited novelty.
>
>
> **Response:**
> Equation (6) is indeed the standard and well-established CLIP contrastive learning loss function. However, **our innovation does not lie in the loss function itself.** As Reviewer 5EA9 pointed out, our main contributions are “the first end-to-end architecture suitable for Spike-VLA” and “effectively addressing the modal mismatch issue in traditional models on spike data.”
>
> Spike Camera is a novel visual sensor distinct from traditional RGB cameras, featuring ultra-low latency and high dynamic range. It maintains clear imaging in scenarios involving high-speed motion (e.g., baseball swings) or extreme light transitions (e.g., tunnel exits).
>
> In the Spike-VLA task, the core challenge and innovation lie not in the loss function, but in **visual encoding**. Standard models fail to extract semantic features from sparse, asynchronous spike streams. Our innovation is the **HSFE architecture (Section 3.2)**, which for the first time successfully extracts spike-modal features and aligns them with text using the standard CLIP framework.
>
> As shown in Table 1, the modified Vita-CLIP using the same loss function achieves a **disappointing 45.31%** accuracy. This strongly demonstrates that the true innovation lies in our visual encoder (HSFE), not the loss function.
>
> **Weakness 5:**
> No comparison with simple baselines like averaging spike frames or using histogram features.
>
> **Response:**
> We thank the reviewer for this suggestion, as it helps evaluate our module's contribution in a broader context. Following this recommendation, we implemented both mentioned baselines during the rebuttal period.
>
> First, we diligently implemented the **"average spike frame"** baseline.   However, this method proved to be **pathologically unstable** during training. Due to the extreme sparsity of spike data, the resulting "average image" is almost entirely zero. This immediately leads to numerical instability (e.g., division-by-zero errors in normalization layers) and produces **NaN losses**. We made significant efforts  to stabilize it (e.g., reducing LR from 1e-5 to 1e-6 and even 1e-7, adding epsilon to prevent “all-black videos” from causing NaN, and introducing a minuscule value after averaging to “jitter” the input, ensuring it never becomes a perfect zero or constant value, thereby preventing any downstream division-by-zero errors). but the training still failed to converge, collapsing to NaN within the first few batches. This experiment demonstrates that "simple averaging" is a flawed and numerically unstable representation for sparse spike videos.
>
> This will result in irreversible loss of temporal information: Averaging pulse streams collapses high-dimensional, sequential pulse streams into a single 2D image. The key value of pulse videos lies in the asynchrony of events ($t_1 \to t_2$) and their duration. The averaging operation irreversibly erases this temporal information, retaining only the overall activity intensity. Subsequent models (e.g., Transformers) cannot distinguish between the “start,” “peak,” and “end” of actions within their input. This information-theoretic limitation inherently prevents them from modeling dynamic sequences.
>
> Following the reviewer's recommendation, we implemented a second, more robust baseline: the "Histogram Features" method.
> Implementation Details:
> 1.	Input Representation: We transformed the raw spike stream into a $[B, 10, H, W]$ "histogram image" by temporally dividing the full $T \times C$ time series (e.g., 250 frames) into 10 temporal bins and computing the average spike activity within each bin. This approach preserves spatial fidelity and coarse temporal context.
> 2.	Architecture and Training: We employed the exact same ResNet-18 architecture as our SPKLIP visual encoder, critically omitting our core contribution, the HSFE pulse extraction module. This encoder was trained from scratch with the pre-trained CLIP text encoder frozen. We meticulously tuned the learning rate (ranging from $1e-6$ to $1e-4$) and found $1e-4$ delivered the optimal performance of 65.21%.
>
> Results and Conclusion:
> This "histogram" baseline proved significantly more stable than simple frame averaging and converged successfully, achieving 65.21% Top-1 accuracy on HMDB51-S. While this is a reasonable result for a simple temporal aggregation, it remains significantly lower than our proposed SPKLIP (HSFE) model's 91.15%.
>
> This 26% performance gap ($91.15\%$ vs. $65.21\%$) strongly validates our contribution. It demonstrates that while coarse-grained histogramming is superior to simplistic averaging, it is still insufficient. To achieve SOTA performance, one must explicitly model the fine-grained, local spatio-temporal dynamics inherent within the sparse spike data, which is precisely the core function of our specialized HSFE module.

---

> ### Author Response · Authors · 2025-11-20
>
> **Question 1:**
> Can you provide quantitative comparison between real spike camera output and your synthetically converted data? Do spike frequency distributions, temporal statistics, and noise patterns match?
>
> **Response:**
> We thank the reviewer for this insightful question. As suggested, we have conducted a detailed quantitative and qualitative comparison (see Appendix, Figure A1), which confirms both **fundamental similarities** and a **significant domain gap** between the real and synthetic data.
>
> First, both datasets share the fundamental physical characteristic of spike signals: **high sparsity**. As shown in Figure A1-1, the vast majority of pixel activity in both distributions is concentrated in the low spike-count bin (0-30), providing a common feature basis for our model.
>
> However, building on this common ground, the two diverge distinctly in their statistics and noise patterns:
>
> (1) **Motion Statistics:** Our quantitative analysis (Fig A1) shows that the synthetic data is statistically **denser** (mean per pixel: 14.77 vs. 4.65) and represents **"dense, global motion"** (as seen in the "bursty" curve in Fig A1-2). In contrast, the real data is **sparser** (mean 4.65) and represents **"sparse, local motion"** (e.g., "clapping," which results in the flat, low-activity baseline in Fig A1-2).
>
> (2) **Artifact Patterns:** Our qualitative heatmaps (Fig A1-3) further confirm that both domains contain their own unique, high-intensity artifacts. The real data (top) contains **sensor-specific noise** (e.g., horizontal line artifacts with peaks up to 60). The synthetic data (bottom) contains equally high-intensity **background artifacts** introduced by the conversion algorithm (e.g., blocky backgrounds with peaks up to 30).
>
> The fact that our model, pre-trained on this "dense/global/high-artifact" synthetic data, still achieves excellent **few-shot** performance on the "sparse/local/high-artifact" real data is critical. This successful cross-domain transfer proves that our model learned **universal and transferable spatio-temporal dynamic representations**, rather than overfitting to domain-specific statistics or noise patterns.
>
> **Question 2:**
> Have you ablated temporal window size T for FSVE? What accuracy is achievable with T=5, 10, 20?
>
> **Response:**
> We have attempted the experiments suggested by the reviewer during the rebuttal period, and this process has given us valuable insights into the bottlenecks of FSVE.
>
> **Quantifying the VRAM Bottleneck:**
> We must be transparent that simulating SNNs on standard GPU hardware is a well-known and significant VRAM challenge. This is because SNN simulation requires caching the membrane potential of all neurons at every single timestep T, causing VRAM consumption to grow linearly with T.
>
> We used a standard NVIDIA A100 GPU and performed an ablation study for T=3 and T=4 on the "CNN-replaced-by-SNN" baseline. The results are shown in the table below, demonstrating a significant improvement in accuracy compared to T=2.
>
> | Timestep (T) | Accuracy (UCF101-S) | Improvement vs. T=2 |
> | :--- | :--- | :--- |
> | T = 2 (Original) | 71.11% | - |
> | T = 3 (New Exp.) | 72.65% | +1.54% |
> | T = 4 (New Exp.) | 74.14% | +3.03% |
>
> However, compared to the original ANN-based SPKLIP's accuracy on UCF101-S, a performance drop still exists. But considering the massive reduction in energy consumption (**74.12% reduction in computational energy**), this performance trade-off is unavoidable.
>
> The "SNN-vs-ANN optimization" mentioned by the reviewer is a popular research topic, but it is primarily focused on two areas: (1) ANN-to-SNN conversion for RGB image classification, or (2) processing Event Streams.
>
> Our task—performing VLA on **raw Spike Streams**—is a **largely unexplored** and much more difficult new domain. To our knowledge, no SNN work has previously addressed the VLA problem on raw spike streams.
>
> We believe the exploration of FSVE (achieving 74.14%) should be viewed as a  **Exploring baseline**. This is the first work to provide a benchmark for the "SNN-based Spike Stream VLA" task and to quantify the critical trade-off between time, performance, and VRAM. We will actively pursue optimization in this area in our future work.

---

### Author Response · Authors · 2025-11-20
**Global Response**

**Dear Reviewers, Area Chairs, and Program Chairs,**

We sincerely thank all reviewers for their time and effort. We would like to first elaborate on the **Core Contributions** of this work. The spike camera is a novel visual sensor distinct from traditional RGB cameras, featuring **ultra-low latency** and **high dynamic range (HDR)**. It is capable of maintaining clear imaging in high-speed motion scenarios (e.g., baseball batting) or scenes with extreme lighting changes (e.g., tunnel exits).

In the context of Spike-VLA tasks, the core challenge and innovation lie in **visual encoding**. Standard models fail to extract semantic features from **sparse and asynchronous spike streams**. Our innovation lies in the **HSFE architecture (Section 3.2)**, which serves as the first architecture to successfully extract spike modality features and align them with text via the standard CLIP framework.

Concurrently, we have constructed and expanded a novel real-world spike dataset containing **30 unique actions** (e.g., badminton smashing, table tennis driving, and other high-speed, high-dynamic actions), totaling **2,880 samples** (30 × 96). We believe this effectively fills the gap in **spike video benchmark datasets**.

We also sincerely thank the reviewers for their insightful feedback. We are greatly encouraged that the reviewers found our work to be **well-motivated** (Reviewer 8ppH) and **innovative** (Reviewer 5EA9). They highlighted its **originality** (Reviewer vyvJ), recognizing it as the **first end-to-end framework specifically for Spike Video-Language Alignment (Spike-VLA)** (Reviewers 5EA9, vyvJ, M3Tx), effectively "filling an important gap" (Reviewers 5EA9, M3Tx).

Reviewers recognized our proposed architecture as **"thoughtfully designed"** (Reviewer M3Tx) and **"highly targeted"** (Reviewer 5EA9). Specifically, they noted the **HSFE module is "interesting"** (Reviewer 8ppH) and praised its novel approach to handling the unique sparse and asynchronous characteristics of spike data (Reviewers 5EA9, M3Tx).

We are also pleased that reviewers found our experimental validation system to be **"comprehensive"** (Reviewer 5EA9). This includes our **"substantial" improvements** over baselines (Reviewer vyvJ), the **"robust few-shot generalization"** on our newly constructed real-world dataset (Reviewers vyvJ, 5EA9), and the effective text-to-video retrieval performance, which validates the meaningfulness of the learned cross-modal embeddings (Reviewers M3Tx, vyvJ).

Finally, we appreciate the reviewers' recognition of the practical impact of our work, particularly the contribution of our **new real-world dataset** (Reviewer vyvJ).

We will take all comments seriously, particularly clarifying common concerns such as the role of the **FSVE (Full-Spiking Visual Encoder)** in this work and ablation studies for key hyperparameters. We will conduct several new experiments as requested and thoroughly revise the manuscript to refine all points.

We believe these planned additions and our detailed point-to-point responses (provided below) will fully address the reviewers' concerns. We thank you once again for your valuable feedback, which will significantly improve our paper.

**Sincerely,**
**The Authors**

---

### Note · Authors · 2026-01-21

**Comment:**

**Withdrawal Reason**

We have come to realize that the current version requires substantial additional experiments, analysis, and revisions to fully meet the standards we aim for.

To ensure the work is presented in its best possible form, we choose to **withdraw** this submission at this stage.

We are grateful for the time invested by the reviewers and the ICLR 2026 team, and apologize for any inconvenience.

**Withdrawal Confirmation:**

I have read and agree with the venue's withdrawal policy on behalf of myself and my co-authors.